# ROBO2 is a stroma suppressor gene in the pancreas and acts via TGF-β signalling

Andreia V. Pinho [1,2,3], Mathias Van Bulck[4], Lorraine Chantrill[1,3,5], Mehreen Arshi[1,3], Tatyana Sklyarova [4], David Herrmann [1,3,5], Claire Vennin[1], David Gallego-Ortega [1,5], Amanda Mawson [1,3], Marc Giry-Laterriere[1,3], Astrid Magenau[1], Gunther Leuckx[6], Luc Baeyens[6], Anthony J. Gill[1,3,7], Phoebe Phillips[8], Paul Timpson[1,3,5], Andrew V. Biankin [3,9,10,11], Jianmin Wu [3,12] & Ilse Rooman[3,4]

Whereas genomic aberrations in the SLIT-ROBO pathway are frequent in pancreatic ductal adenocarcinoma (PDAC), their function in the pancreas is unclear. Here we report that in pancreatitis and PDAC mouse models, epithelial Robo2 expression is lost while Robo1 expression becomes most prominent in the stroma. Cell cultures of mice with loss of epithelial Robo2 (Pdx1$^{Cre}$;Robo2$^{F/F}$) show increased activation of Robo1$^+$ myofibroblasts and induction of TGF-β and Wnt pathways. During pancreatitis, Pdx1$^{Cre}$;Robo2$^{F/F}$ mice present enhanced myofibroblast activation, collagen crosslinking, T-cell infiltration and tumorigenic immune markers. The TGF-β inhibitor galunisertib suppresses these effects. In PDAC patients, ROBO2 expression is overall low while ROBO1 is variably expressed in epithelium and high in stroma. ROBO2$^{low}$;ROBO1$^{high}$ patients present the poorest survival. In conclusion, Robo2 acts non-autonomously as a stroma suppressor gene by restraining myofibroblast activation and T-cell infiltration. ROBO1/2 expression in PDAC patients may guide therapy with TGF-β inhibitors or other stroma /immune modulating agents.

[1] Cancer Division, The Garvan Institute of Medical Research, Sydney Darlinghurst 2010 NSW, Australia. [2] Faculty of Medicine and Health Sciences, Macquarie University, Sydney Macquarie University 2109 NSW, Australia. [3] Australian Pancreatic Cancer Genome Initiative (APGI), Sydney Darlinghurst 2010 NSW, Australia. [4] Oncology Research Centre, Vrije Universiteit Brussel, Brussels 1090, Belgium. [5] St. Vincent's Clinical School, UNSW, Sydney Darlinghurst 2010 NSW, Australia. [6] Beta cell Neogenesis Lab, Vrije Universiteit Brussel, Brussels 1090, Belgium. [7] Cancer Diagnosis and Pathology Group, Kolling Institute of Medical Research, Royal North Shore Hospital, University of Sydney, Sydney St. Leonards 2065 NSW, Australia. [8] Lowy Cancer Research Centre, University of New South Wales, Sydney Sydney 2052 NSW, Australia. [9] Wolfson Wohl Cancer Research Centre, Institute of Cancer Sciences, University of Glasgow, Glasgow G61 1BD Scotland, UK. [10] West of Scotland Pancreatic Unit, Glasgow Royal Infirmary, Glasgow G5 0SF Scotland, UK. [11] South Western Sydney Clinical School, UNSW, Liverpool Liverpool 2170 NSW, Australia. [12] Key laboratory of Carcinogenesis and Translational Research (Ministry of Education/Beijing), Center for Cancer Bioinformatics, Peking University Cancer Hospital & Institute, Beijing 100142, China. These authors contributed equally: Andreia V. Pinho, Mathias Van Bulck. Correspondence and requests for materials should be addressed to A.V.P. (email: andreia.pinho@mq.edu.au) or to I.R. (email: irooman@vub.be)

Pancreatic ductal adenocarcinoma (PDAC) has a dismal prognosis, being predicted to become the second leading cause of cancer-related death by 2030[1]. Chronic pancreatitis is a major risk factor for the development of PDAC. Both diseases share a common aetiology in the exocrine pancreatic tissue and are characterised by a strong desmoplastic response, comprised of multiple stromal cell types, including activated myofibroblasts or stellate cells and immune cell infiltrates[2,3].

Next-generation sequencing technology transformed our understanding of the genetic alterations associated with the genesis and progression of PDAC. Within a very heterogeneous mutational landscape, specific signalling pathways and biological mechanisms are recurrently affected, including SLIT ligands and ROBO receptors, originally described for their role in axon guidance[4]. SLIT–ROBO genes present structural variations or mutations in approximately one-third of patients[5–7] and are also frequently silenced by methylation[8]. Analyzing whole-tissue gene expression, we previously showed a progressive loss of Robo2, a progressive increase in Robo1 and no change in Robo3 mRNA expression when sequentially comparing normal mouse pancreas, models of acinar-to-ductal metaplasia and a genetically engineered mouse model of PDAC[5]. In addition, low expression of ROBO2 mRNA was associated with poor patient survival[5].

Studies focusing on the role of the SLIT–ROBO signalling pathway in the exocrine pancreas have, however, been very limited. Loss-of-function experiments in tumour cells showed that ROBO1 and ROBO3 receptors are involved in pancreatic cancer cell migration and metastasis[9,10], and SLIT2 has a role in neural remodelling associated with PDAC[11,12].

In this study, we describe cell-type-specific changes in Robo and Slit gene expression during pancreatitis and tumour development, and we reveal their functional consequence in reshaping the tissue's constituents. More specifically, our data show that overall Robo2 is lost in PDAC. Loss of Robo2 signalling in the exocrine epithelium reprograms the microenvironment, resulting in the prominent activation of myofibroblasts and increased T-cell infiltration, with a critical role for transforming growth factor beta (TGF-β) signalling. Our results provide insights into the mechanisms of the desmoplastic response characteristic of exocrine pancreatic pathologies, which is the subject of investigation in clinical trials.

## Results

### Robo1 and 2 receptors show cell-type-specific changes.
We assessed Robo1 and Robo2 expression by RNA in situ hybridization (RISH), within the exocrine tissue of normal mouse pancreas (NMP), acute pancreatitis (AP) induced by caerulein treatment[13], and PDAC lesions of Kras[G12D]; Trp53[R172H]; Pdx1[Cre] (KPC) animals[14]. Expression in the endocrine tissue is shown in the Supplementary material (Supplementary Figure 1a).

In NMP, Robo1 and Robo2 are expressed within the acinar (Ac) and ductal (Du) epithelium, as well as in the non-epithelial (Fig. 1a, f), vimentin-positive (Vim+) mesenchymal (Me) cells (Fig. 1f).

Robo2 expression is decreased in the epithelial acinar cells in caerulein-induced AP and in KPC, to the extent that the RISH signal is close to zero (Fig. 1b, c, e). In these conditions, Robo1 is increased in Vim+ and α-smooth muscle actin-positive (αSma/Acta2+) mesenchymal cells (Fig. 1b, c, d, g and Supplementary Figure 1b), characteristic of activated myofibroblasts. Robo1 remains detectable in the epithelial lesions of the KPC mice (Fig. 1c, d, g).

In conclusion, in diseases of the exocrine pancreas, epithelial Robo2 expression is lost, while Robo1 expression is increased in the mesenchyme.

### Epithelial Robo2 restrains activation of myofibroblasts.
Since ROBO2 is frequently mutated in pancreatic tumour epithelium[5] and epithelial gene expression is suppressed in pancreatitis and PDAC (Fig. 1), we genetically inhibited Robo2 in pancreatic epithelium. This was accomplished by breeding Robo2 exon 5 conditional knockout animals with the Pdx1[Cre] strain (from here on referred to as Robo2[F/F]) that resulted in an 80% reduction in total pancreatic Robo2 mRNA compared with controls (Supplementary Figure 2a). In the Robo2-depleted epithelial acinar and duct cells, we noticed a slight, and possibly compensatory, upregulation of Robo1 expression (Supplementary Figure 2b). Adult Robo2[F/F] animals presented normal pancreas histology (Supplementary Figure 2c) with endocrine hormone expressing islets (Supplementary Figure 2d) and amylase expressing exocrine tissue comparable with controls (Supplementary Figure 2d, e). We observed no changes in exocrine (Supplementary Figure 2f) or endocrine gene expression (Supplementary Figure 2g), except a subtle increase in SRY-Box9 (Sox9) (Supplementary Figure 2f). Robo2[F/F] animals did not show changes in the mesenchymal markers snail family transcriptional repressor 1 (Snai1), Vim or the myofibroblast marker Acta2 (Supplementary Figure 2h).

An established in vitro assay with highly enriched exocrine cell fractions mimics the epithelial cell changes (acinar to ductal metaplasia) that occur during pancreatitis[15–18]. Using cells from the Pdx1[Cre] mice in this assay (left panels in Fig. 2a, c, e), we observed changes similar to our previous work[15–18], resulting in epithelial monolayers. In contrast, more cells from Robo2[F/F] pancreata attached to the culture plate at 48 h (Fig. 2b), with mostly mesenchymal cells growing around few epithelial colonies that were not as spread out as in the controls (Fig. 2a, c, e right panels).

At day 8 (D8), overall expression of the epithelial cell markers pancreas transcription factor 1a (Ptf1a), carboxypeptidase1 (Cpa1), pancreas and duodenal homeobox gene 1 (Pdx1) and keratin19 (Krt19) was decreased in Robo2[F/F] cell cultures (Fig. 2d), whereas expression of mesenchymal genes Snai1, twist family BHLH transcription factor 1 (Twist1) and Vim and the activated myofibroblast genes Acta2 and desmin (Des) were upregulated (Fig. 2f). The latter was confirmed by staining for Vim and αSma (Fig. 2e). We note that both in Pdx1[Cre] as in Robo2[F/F], the Vim+ cells express αSma. In Robo2[F/F] cultures, E-cadherin (E-cad)+ monolayer formation was low and their E-cad staining pattern was punctuated, indicative of immature adherens junctions (Fig. 2c, right panel). Double-positive E-cad/Vim or E-cad/αSma cells were never found.

To further characterise the origin of the fibroblastic cell population, we used fluorescence-activated cell sorting (FACS) to separate the epithelial (EpCAM+, Cd31− and Cd140a/Pdgfr-α−) and mesenchymal (Cd140a/Pdgfr-α+, Cd31− and EpCAM−) cell populations (Supplementary Figure 3a)[19]. Specificity of EpCAM and Cd140a markers was confirmed by co-staining with E-cad and Vim (Supplementary Figure 3b). This experiment yielded more mesenchymal cells from the Robo2[F/F] D8 cell cultures, confirming the above results (Fig. 2g). In this experiment, we also analysed the presence of Cre-mediated recombination in the Robo2 allele (deletion of Robo2 exon 5). The 1180 base-pair (bp) recombined (R) band that we find in the epithelial Robo2[F/F] population, is not retrieved in the mesenchymal Robo2[F/F] population (Fig. 2h). The Robo2[F/F] mesenchymal cells show a similar band to wild-type epithelial Pdx1[Cre] cells, i.e. a non-recombined (N) band of 1390 bp (Fig. 2h). The genotype of Pdx1[Cre] and Robo[F/F] cells was confirmed via restriction digestion of the respective amplicon (Supplementary Figure 3c). This led us to conclude that the mesenchymal cells present in Robo2[F/F] cultures are not of epithelial origin and get activated in a non-autonomous manner, which leads to their expansion in culture.

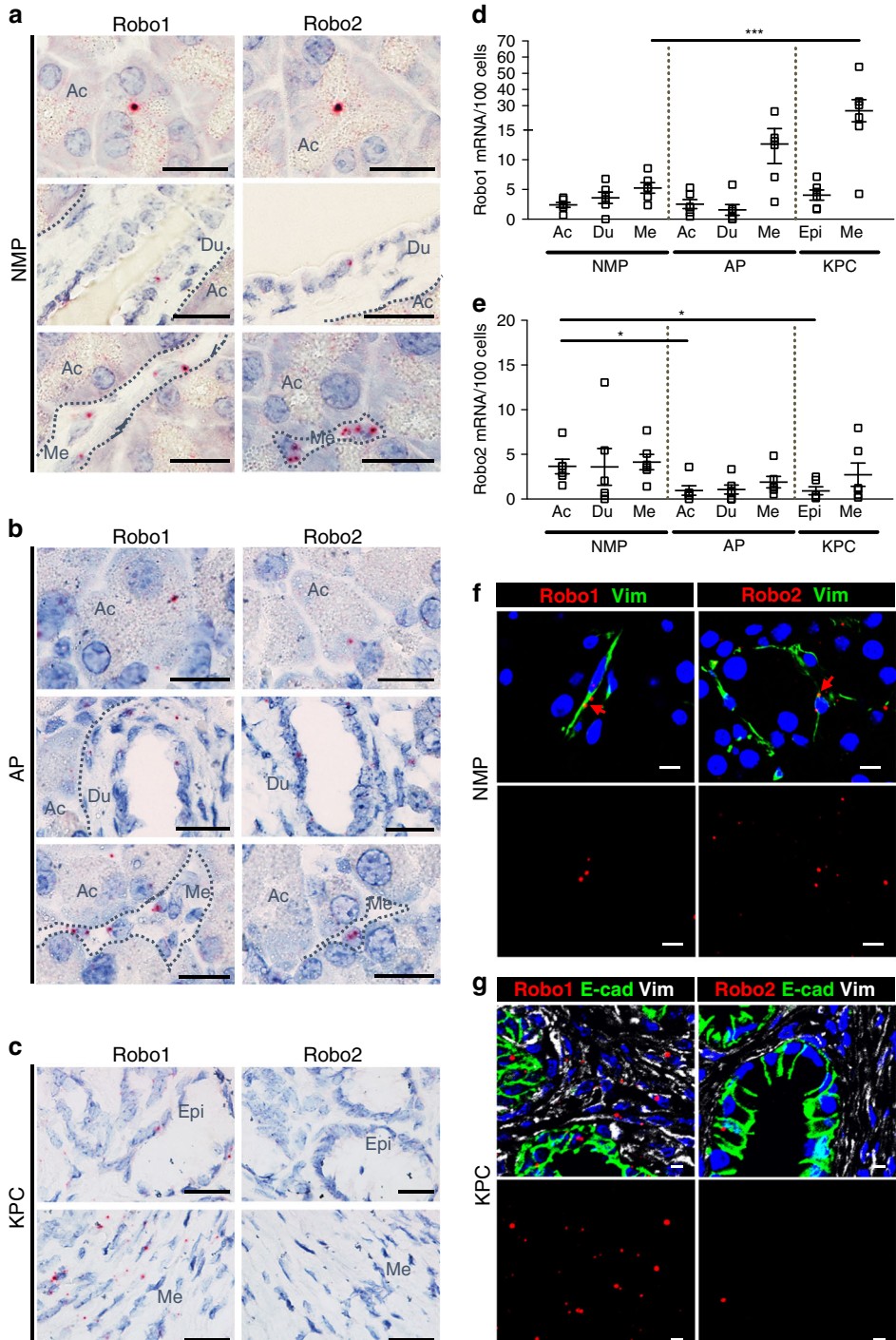

**Fig. 1** Cell-type-specific changes in Robo1/2 expression in pancreatitis and PDAC. **a–c** Robo1 and Robo2 mRNA expression analysed by RNA in situ hybridization (RISH) in **a** normal mouse pancreas (NMP), **b** acute pancreatitis (AP) and **c** PDAC samples from Kras$^{G12D}$; Trp53$^{R172H}$; Pdx1$^{Cre}$ (KPC) animals. Note that dot intensity is not related to the amount of mRNA copies. Dotted lines delineate histological compartments (acini, Ac; ducts, Du; tumour epithelium, Epi; mesenchymal cells, Me). Images are representative of six independent experiments. Scale bars correspond to 20 μm. **d** Quantification of Robo1 RISH in tissue sections of NMP, AP and PDAC. Data presented as mean $+/-$ SEM; $N = 6$. Statistical analysis was performed using an unpaired $t$ test with Welsh's correction; ***$P < 0.001$. **e** Quantification of Robo2 RISH in tissue sections of NMP, AP and PDAC. Data presented as mean $+/-$ SEM; N $= 6$. Statistical analysis was performed using an unpaired $t$ test with Welsh's correction; *P $< 0.05$. **f** RISH— immunofluorescence multiplexing of Robo1 and Robo2 with the epithelial marker E-cad and the mesenchymal marker Vim in NMP. Red arrows indicate Robo1$^+$ and Robo2$^+$ signal. Nuclei are stained with DAPI. Images are representative of six independent experiments. Scale bars correspond to 20 μm. Confocal pictures were acquired using 20x magnification with zoom 3.5. **g** RISH—immunofluorescence multiplexing of Robo1 or Robo2 with the epithelial marker E-cadherin (E-cad) and the mesenchymal marker vimentin (Vim) in PDAC (KPC model). Nuclei are stained with DAPI. Images are representative of three independent experiments. Scale bars correspond to 20 μm. Confocal pictures were acquired using 20x magnification with zoom 2.0

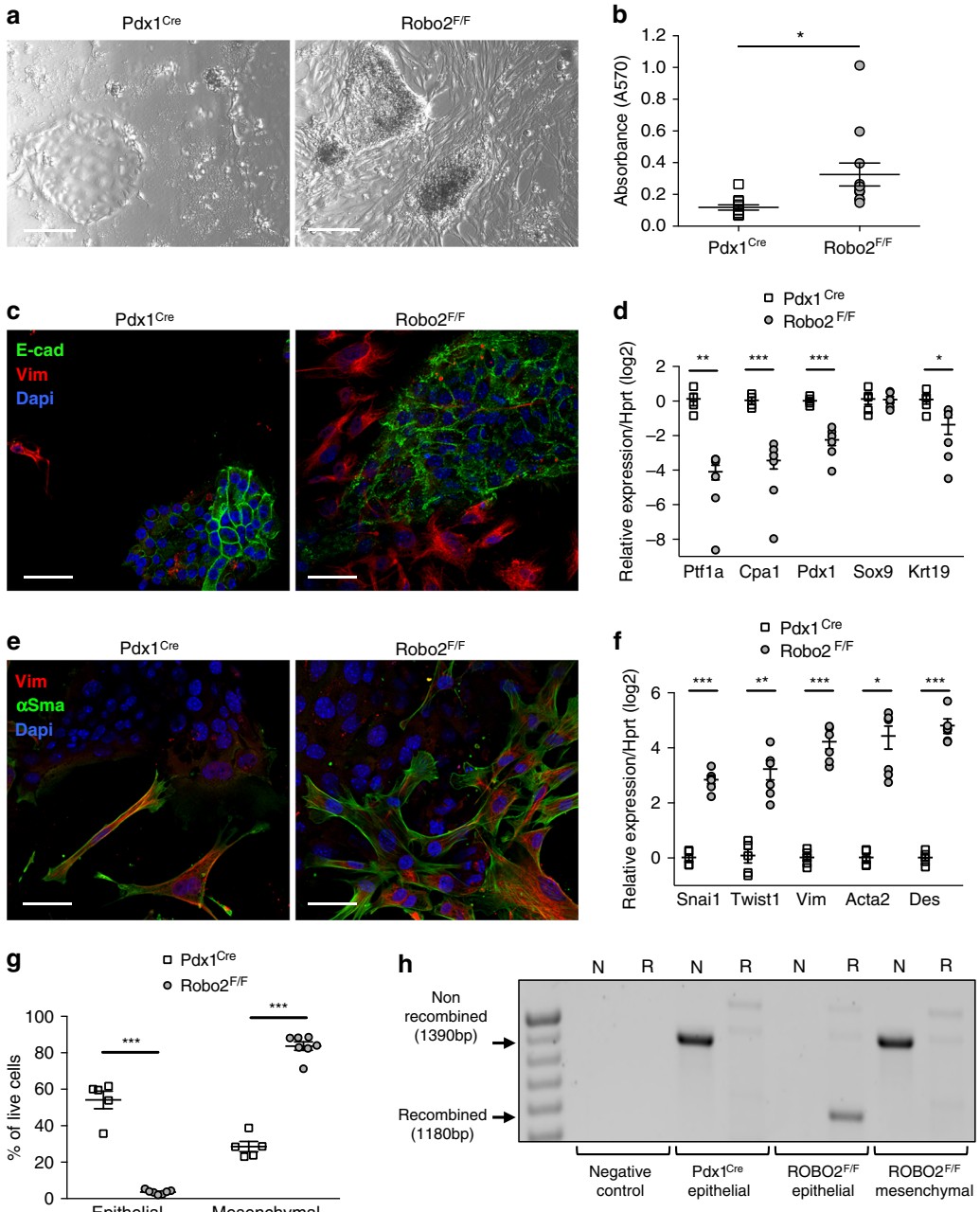

**Fig. 2** Increased myofibroblasts in cell cultures from Robo2[F/F] pancreas. **a** Representative images of Pdx1[Cre] and Robo2[F/F] primary pancreatic exocrine cultures at day 8 (D8) of culture. Scale bars correspond to 200 μm. **b** Quantification of sulphorhodamine B staining as a measurement of cell attachment at 48 h of culture. **c** Immunofluorescence staining of the mesenchymal marker Vim and the epithelial marker E-cad in D8 primary pancreatic cultures of Pdx1[Cre] and Robo2[F/F] animals. Images are representative of five independent experiments. Nuclei are stained with DAPI. Confocal images were acquired using 40x magnification. Scale bars correspond to 50 μm. **d** mRNA expression of pancreatic epithelial markers analysed by RT-qPCR in D8 primary pancreatic cultures of Pdx1[Cre] and Robo2[F/F] animals. **e** Immunofluorescence staining of the mesenchymal markers Vim and αSma in D8 primary pancreatic cultures of Pdx1[Cre] and Robo2[F/F] animals. Images are representative of five independent experiments. Nuclei are stained with DAPI. Confocal images were acquired using 40x magnification. Scale bars correspond to 50 μm. **f** mRNA expression of mesenchymal and stellate cell markers analysed by RT-qPCR in D8 primary pancreatic cultures of Pdx1[Cre] and Robo2[F/F] animals. All qPCR data are referred to housekeeping gene Hprt. **g** Quantification of FACS-sorted epithelial (EpCAM[+], Cd31[−] and Cd140a[−]) and mesenchymal (Cd140a[+], Cd31[−] and EpCAM[−]) cell populations in D8 primary pancreatic cultures of Pdx1[Cre] and Robo2[F/F] animals. **h** Analysis of Cre-mediated recombination of the Robo2 allele by PCR using 10 ng of genomic DNA from FACS-sorted D8 primary pancreatic cultures of Pdx1[Cre] and Robo2[F/F] animals. Arrows indicate the non-recombined (1390 bp) and recombined (1180 bp) amplicons. N or R indicates which primer pair was used. N for Robo2 wild-type allele primer pair, R for Robo2 knockout allele primer pair. All data presented as mean $+/-$ SEM; $N \geq 5$. Statistical analysis was performed using unpaired $t$ test with Welch's correction; $*P < 0.05$, $**P < 0.01$, $***P < 0.001$

**Robo2-mediated myofibroblast activation depends on TGF-β.**
We hypothesised that Wnt signalling, a known pathway regulated by Slit2–Robo1[20–22], and/or TGF-β signalling, known for myofibroblast cell activation[23], were implicated in our observations.

Indeed, in Robo2[F/F] cell cultures, we found upregulation of the Wnt signalling genes Axin2, Lef1 (lymphoid enhancer binding factor 1), Lgr5 (leucine-rich repeat containing G-protein-coupled receptor 5) and Mmp9 (matrix metalloproteinase 9) (Fig. 3a). We

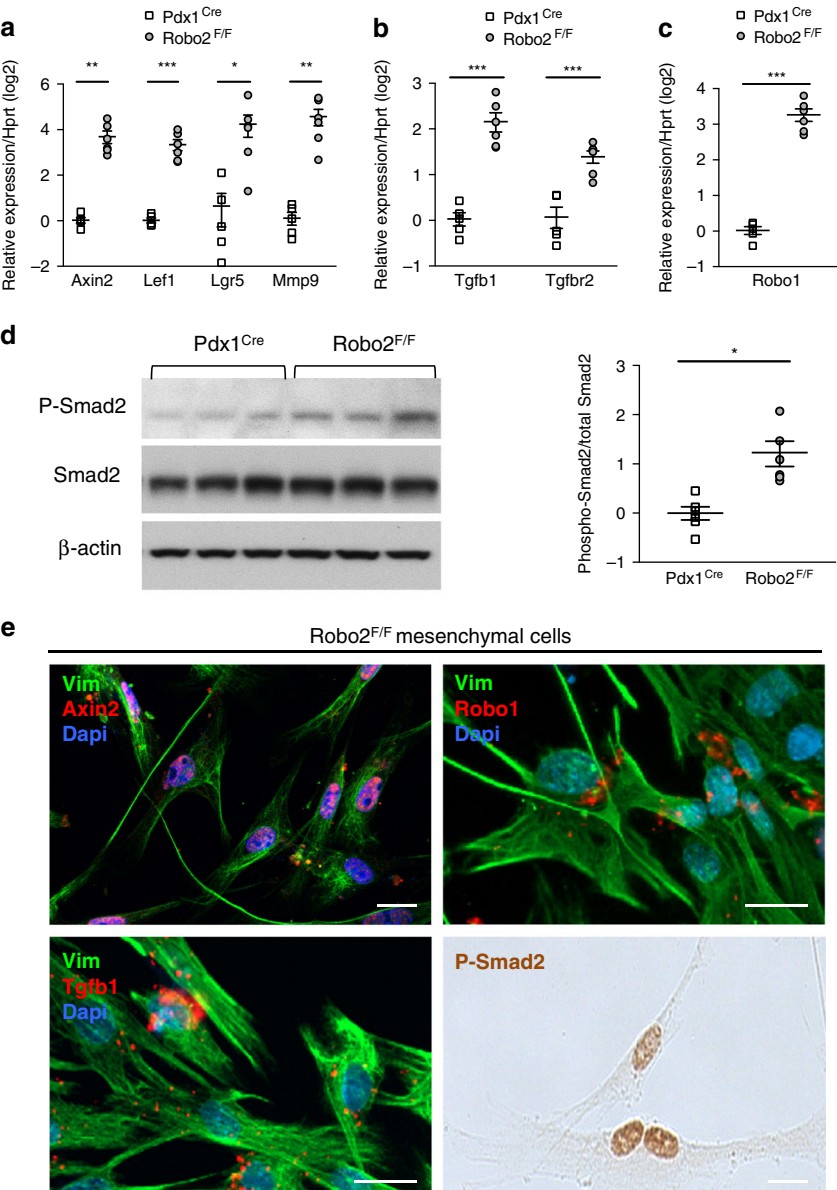

**Fig. 3** Activation of Wnt and TGF-β pathway in cell cultures from Robo2[F/F] pancreas. **a**, **b**, **c** mRNA expression of Wnt targets (**a**), Tgfb1, Tgfbr2 (**b**) and Robo1 (**c**) analysed by RT-qPCR in D8 primary pancreatic cultures of Pdx1[Cre] and Robo2[F/F] animals. All qPCR data are referred to housekeeping gene Hprt. **d** Western blot (WB) analysis of phospho-Smad2 in D8 cultures and quantification of WB band density using ImageJ. **e** Representative images of mesenchymal cells, identified by Vim, in D8 Robo2[F/F] cultures stained with immunofluorescence/RISH for Axin2, Tgfb1, Robo1 or phospho-Smad2. Nuclei in immunofluorescence images are stained with DAPI. Images are representative of three to four independent experiments. Scale bars correspond to 20 μm. All data presented as mean +/− SEM; $N = 4$-6. Statistical analysis was performed using unpaired $t$ test with Welch's correction; *$P < 0.05$, **$P < 0.01$, ***$P < 0.001$

also detected increased mRNA of TGF-β ligand (Tgfb1) and TGF-β receptor II (Tgfbr2) (Fig. 3b), as well as increased phosphorylation of Smad2 (Fig. 3d), a hallmark of TGF-β pathway activation. Robo1 expression was upregulated (Fig. 3c), consistent with being a target of Wnt and TGF-β signalling pathway[24,25].

The Robo2[F/F] cell cultures contain a majority of myofibroblasts, and these cells stain positive for Axin2, Tgfb1, Robo1 and phospho-Smad2 (P-Smad2) (Fig. 3e), hence they contribute to the increases detected by RT-PCR and underscore the non-autonomous effect of epithelial Robo2 deletion.

The TGF-β receptor I inhibitor galunisertib has recently shown promising results in a randomised phase 2 clinical trial for advanced PDAC and is further under study[26,27]. Robo2[F/F] cultures treated with galunisertib have decreased phospho-Smad2 (Supplementary Figure 4a). In Robo2[F/F] cell cultures

treated with galunisertib, there was an inhibition of the effects seen in untreated Robo2[F/F] cells, i.e. epithelial markers E-cad, Krt19 and Cpa1 were upregulated, and mesenchymal markers Vim, Snail, Acta2/αSma as well as Tgfb1, Tgfbr2 or Robo1 were downregulated, compared with Robo2[F/F] controls (Fig. 4a–c). Changes in αSma, Vim and E-cad were confirmed by western blot (Supplementary Figure 4b) and immunofluorescence (Fig. 4d). We observed that in the mesenchymal cells of the Robo2[F/F] cultures, the expression of αSma as well as that of Robo1 was suppressed by galunisertib (Fig. 4e, f).

In addition, we found that galunisertib reduced the Tgfb1 increase found in Robo2[F/F] epithelial cells (Supplementary Figure 5a, b). The fact that Tgfb1 increased in Robo2[F/F] epithelial cells was further confirmed by ROBO2 knockdown using siRNA in Panc1 cells—chosen because it is an epithelial tumour cell line

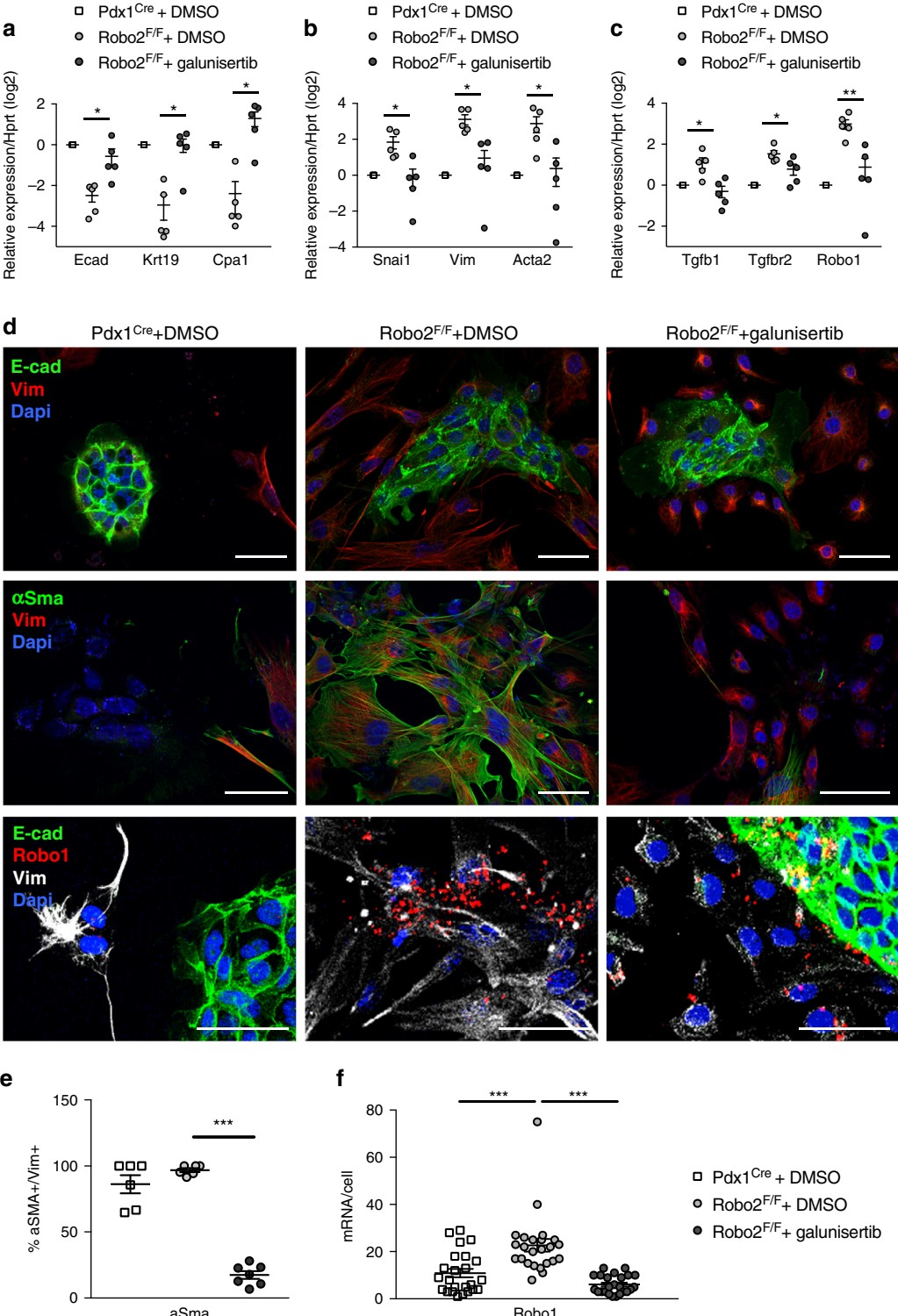

**Fig. 4** Myofibroblasts activation in Robo2$^{F/F}$ cell cultures depends on TGF-β. **a–c** mRNA expression of epithelial (**a**) and mesenchymal (**b**) markers and TGF-β pathway genes (**c**) in D8 primary pancreatic cultures, treated with galunisertib or DMSO vehicle. All data referred to Hprt and presented as mean +/− SEM, N = 5. Statistical analysis was performed using unpaired $t$ test with Welch's correction; *$P < 0.05$, **$P < 0.01$. **d** Immunofluorescence staining and RISH for Robo1 with epithelial marker E-cad and the mesenchymal markers Vim and/or αSma in D8 pancreatic cultures treated with galunisertib or DMSO vehicle. Nuclei are stained with DAPI and confocal images were acquired using 40x magnification. Images are representative of three to five independent experiments. Scale bars correspond to 50 μm. **e** Quantification of the percentage of activated myofibroblasts (αSma$^+$; Vim$^+$) in total Vim$^+$ mesenchymal cells in D8 primary pancreatic cultures, treated with galunisertib or DMSO vehicle. **f** Quantification of Robo1 mRNA by RISH per mesenchymal cell in D8 primary pancreatic cultures, treated with galunisertib or DMSO vehicle. Data presented as mean +/− SEM; data from $N \geq 3$ mice per group. Statistical analysis was performed using unpaired $t$ test with Welsh's correction; ***$P < 0.001$

that retains some ROBO2 expression. ROBO2 knockdown led to an upregulation of TGFB1 ligand, whereas no change in TGFBR2, Wnt signalling or ROBO1 was detected (Supplementary Figure 5c).

We conclude that TGF-β pathway inhibition blocks the myofibroblast activation induced in conditions of epithelial Robo2 loss.

**Robo2 suppresses stromal activation in pancreatitis.** We verified if the effects observed in the in vitro model are reproduced in AP. Samples were collected at day 3 (D3-acute damage) and at day 8 (D8-regeneration) (Fig. 5a). At the histological level, we noticed that Robo2[F/F] pancreata had more inter-acinar space at D3 (Fig. 5a), indicative of stromal changes. Similar to our in vitro observations, RT-qPCR analysis at D3 showed in Robo2[F/F] a lower expression of epithelial genes, a higher expression of desmin and an upregulation of Tgfb1, Tgfbr2 and Robo1 (Fig. 5b, c, d). Increased activation of the TGF-β pathway in Robo2[F/F] animals was confirmed by phospho-Smad2 IHC (Fig. 6a, d), as well as the increase in the number of αSma[+] myofibroblasts (Fig. 6b, e). There was no increase in acinar-to-ductal metaplasia (Supplementary Figure 6a,b) and E-cad localization at the cell membrane or E-cad mobility[28], did not differ between Pdx1[Cre] and Robo2[F/F] pancreas (Supplementary Figure 7). At D8, the epithelial tissue had also regenerated to the same extent as the Pdx1[Cre] samples (Fig. 5a). In contrast, the stromal compartment remained different with increased deposition of collagen (Supplementary Figure 6c,d) and increased collagen crosslinking at D8, quantified by second-harmonic generation (SHG) microscopy analysis (Fig. 5h, i). We therefore conclude that, similar to in vitro observations, the Robo2-mediated effects were stroma-specific.

Immune cells are important mediators of the epithelial and stromal changes in pancreatitis and in PDAC[2,29]. Hence, we also characterised the immune cell component on the acute damage time point. Immunohistochemistry showed no difference in the number of F4/80[+] macrophages (Supplementary Figure 6e, f), whereas the number of CD3[+] T lymphocytes was increased in Robo2[F/F] animals compared with Pdx1[Cre] controls (Fig. 6c, f). Targeted qPCR analysis showed increased expression of the anti-inflammatory cytokine interleukin-10 (Il10), but no difference in expression of the pro-inflammatory cytokines Il1b and Il6 (Fig. 5e). Additionally, pathway-focused gene expression analysis, using the Mouse Cancer Inflammation and Immunity Crosstalk RT[2] Profiler PCR Arrays (Qiagen) containing 84 genes involved in inflammation and immunity (Supplementary Table 1) showed downregulation of the pro-inflammatory cytokine Il1a and upregulation of the chemokine receptors Ccr5 and Cxcr2 (Fig. 5f, g). Ccr5 and Cxcr2 promote pancreatic cancer progression[30,31] and metastasis (in the case of Cxcr2), and their chemical inhibition has been proposed as cancer immunotherapy[32,33]. Additionally, we found an upregulation of Toll-like receptor 7 (Tlr7), also previously implicated in pancreatic carcinogenesis[34], and of the tumour suppressor Trp53. Tlr7 regulates the expression of Trp53 itself, as well as of TGF-β and Mmp9 in pancreatic cells[34,35]. Altogether, these results show that loss of epithelial Robo2 leads to an exacerbated immune response that is of anti-inflammatory nature and has been previously linked with PDAC progression and metastasis.

FACS analysis of pancreatic tissue of Robo2[F/F] animals at D3 AP (Supplementary Figure 8a-c) allowed us to assess expression of the most relevant genes per cell type; Axin2 and Robo1 were most prominent in mesenchymal cells, while the TGF-β ligand was expressed in epithelial, stromal and immune cells (Supplementary Figure 8b), as confirmed by RISH (Supplementary Figure 8c).

To determine whether inhibition of the TGF-β pathway in vivo could inhibit Robo2-mediated stromal effects observed after pancreatitis, we administered galunisertib during the caerulein treatment (Fig. 6). This prevented the increase in phospho-Smad2 observed in Robo2[F/F] animals (Fig. 6a, d), evidencing that the drug inhibits TGF-β signalling.

Galunisertib decreased the accumulation of αSma[+] myofibroblasts in Robo2[F/F] (Fig. 6b, e) as well as the infiltration of CD3[+] T lymphocytes (Fig. 6c, f).

Together, the in vivo data show that loss of Robo2 specifically in the epithelial compartment leads to activation and remodelling of the stroma, affecting mesenchymal and immune cells. The effects are mediated by TGF-β signalling and chemical inhibition of this pathway using galunisertib could suppress myofibroblast activation and CD3[+] T lymphocyte accumulation.

**No effect of additional Robo2 loss in KC mice.** As shown in Fig. 1, epithelial Robo2 expression is lost in different conditions of Ras activation, such as AP, where wild-type Ras is overactivated[17,36] and in the KPC mice[14] that express oncogenic KRas[G12D]. We crossed the KC model with Robo2[F/F] animals and treated these animals with caerulein to induce a chronic pancreatitis (Supplementary Figure 9a). Downregulation of Robo2 expression was confirmed in KC animals, which express oncogenic Kras[G12D] in the pancreas (Supplementary Figure 9b). The area occupied by neoplastic lesions was similar between KC_Robo2[F/F] and KC controls, with both groups presenting a mix of early-stage lesions (PanIN1A/B and PanIN2) (Supplementary Figure 9c, d). We also did not find any additional increase in αSma[+] cells, CD3[+] T-cell infiltration or higher TGF-β RNA expression (Supplementary Figure 9e–j).

In conclusion, Robo2 knockout in the KC mouse background does not increase the typical phenotypic changes in epithelium and stroma. One explanation for this observation is the fact that Kras[G12D] mutation on its own already induced Robo2 loss.

**Slit1 suppresses stromal remodelling in pancreatitis.** Slit1 and Slit2 ligands are mainly expressed in the mesenchymal cells of the pancreas (Supplementary Figure 10a, e) and increased in AP and KPC (Supplementary Figure 10b, c, e and f). Since Slit2 knockout animals are not viable[37], we proceeded studying Slit1 knockout (Slit1[−/−]) mice.

The adult pancreas of the Slit1[−/−] mice presented normal histology (Supplementary Figure 12b). Slit1[−/−] pancreatic cell explants showed a similar phenotype to Robo2[F/F] cells with increased cell attachment (Supplementary Figure 11a, b), and increased expression of mesenchymal genes, including αSma/Acta2, as well as increased expression of Wnt signalling genes (Supplementary Figure 11c-e). Although expression changes in Slit1[−/−] were similar to those in Robo2[F/F] cultures (Figs. 2f, 3a) no increase in Tgfb1 or Tgfbr2 was found (Supplementary Figure 11f). When subjected to AP, Slit1[−/−] pancreata showed an increase in inter-acinar space with more αSma[+] myofibroblasts (Supplementary Figure 12a–d) and increased collagen cross-linking at D8 (Supplementary Figure 12c, f). CD3[+] T-lymphocyte infiltration only showed a tendency for increase (Supplementary Figure 12c, e).

We conclude that the absence of Slit1 in all pancreatic cells (i.e. epithelium and stroma) results in enhanced myofibroblast activation in AP or its equivalent cell culture model. Albeit that additional different mechanisms may have been involved, the above observations strongly support that the ligand Slit1 also suppresses the microenvironment by restraining myofibroblast activation.

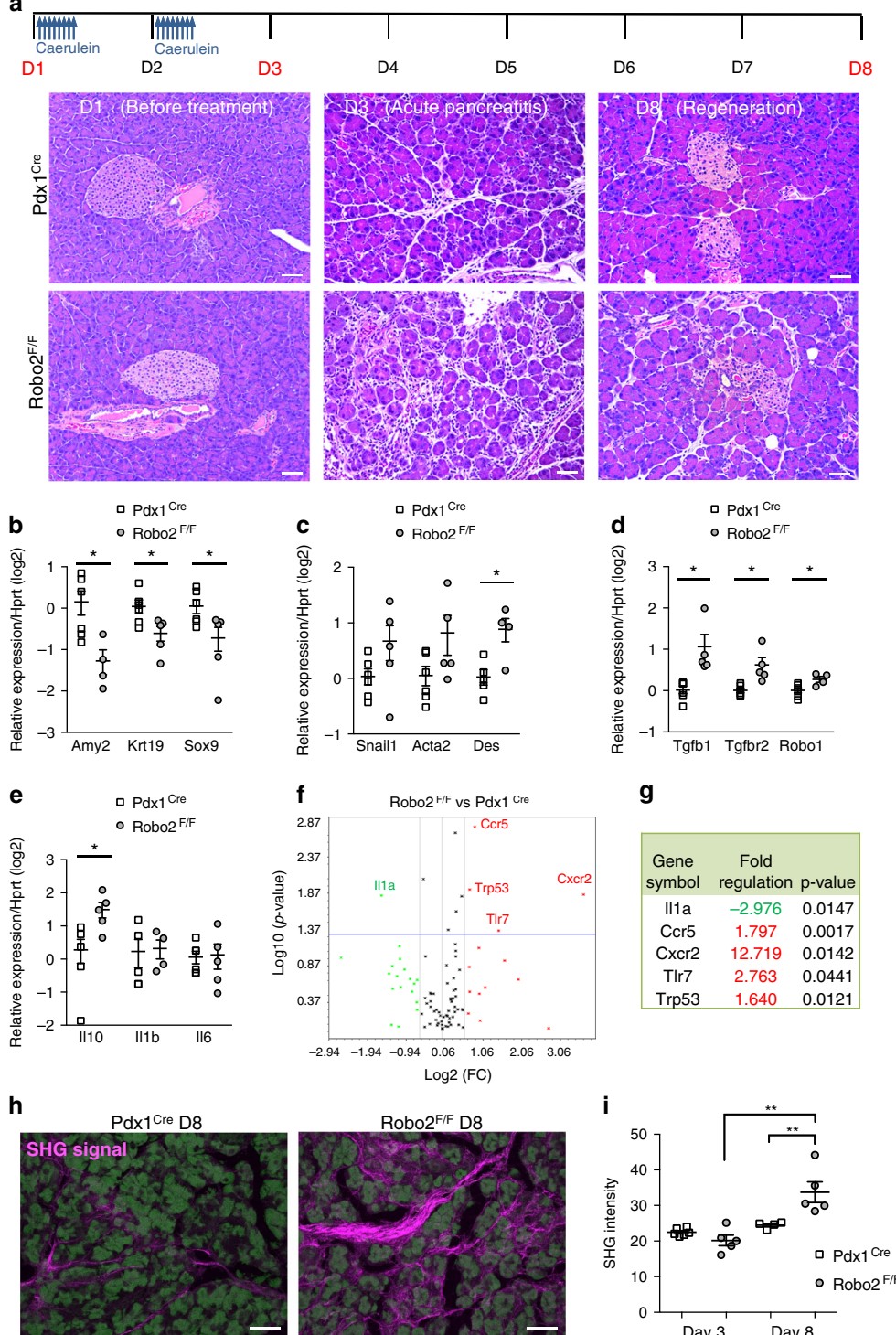

**Fig. 5** Activation of stromal and immune cells during pancreatitis in Robo2$^{F/F}$ mice. **a** Acute pancreatitis (AP) was induced in Pdx1$^{Cre}$ and Robo2$^{F/F}$ animals by intraperitoneal administration of caerulein during 2 consecutive days and pancreata were analysed 3 days (D3) and 8 days (D8) after initiation of treatment. Haematoxylin and eosin (H&E) staining of pancreas sections at D3 and D8 of AP. Scale bars correspond to 50 μm. Images are representative of five to six independent experiments. **b-e** mRNA expression of epithelial (**b**), mesenchymal (**c**), TGF-β pathway genes (**d**) and immune (**e**) markers at D3 AP in Pdx1$^{Cre}$ and Robo2$^{F/F}$ animals. All qPCR data are referred to housekeeping gene Hprt. Data presented as mean +/− SEM; $N = 4$–6. Statistical analysis was performed using unpaired $t$ test with Welch's correction, *$P < 0.05$. **f** Volcano plot representing gene expression of immune markers in total pancreas of Pdx1$^{Cre}$ and Robo2$^{F/F}$ animals at D3 AP, analysed using the RT2 Profiler PCR array Mouse Cancer Inflammation & Immunity Crosstalk (Qiagen). **g** List of genes with differential expression between Pdx1$^{Cre}$ and Robo2$^{F/F}$ animals at D3 AP (fold change > 1.5 and $p < 0.05$). **h** Representative maximum projections of second-harmonic generation (SHG) signal (purple) derived from pancreata sections at D8 of AP. Autofluorescence of pancreatic tissue is represented in green. Scale bars correspond to 50 μm. **i** Quantification of second-harmonic generation (SHG) signal intensity to determine collagen crosslinking. Data presented as mean +/− SEM; $N = 4$–6. Statistical analysis was performed using one-way ANOVA with Turkey's multiple comparisons test, **$P < 0.01$

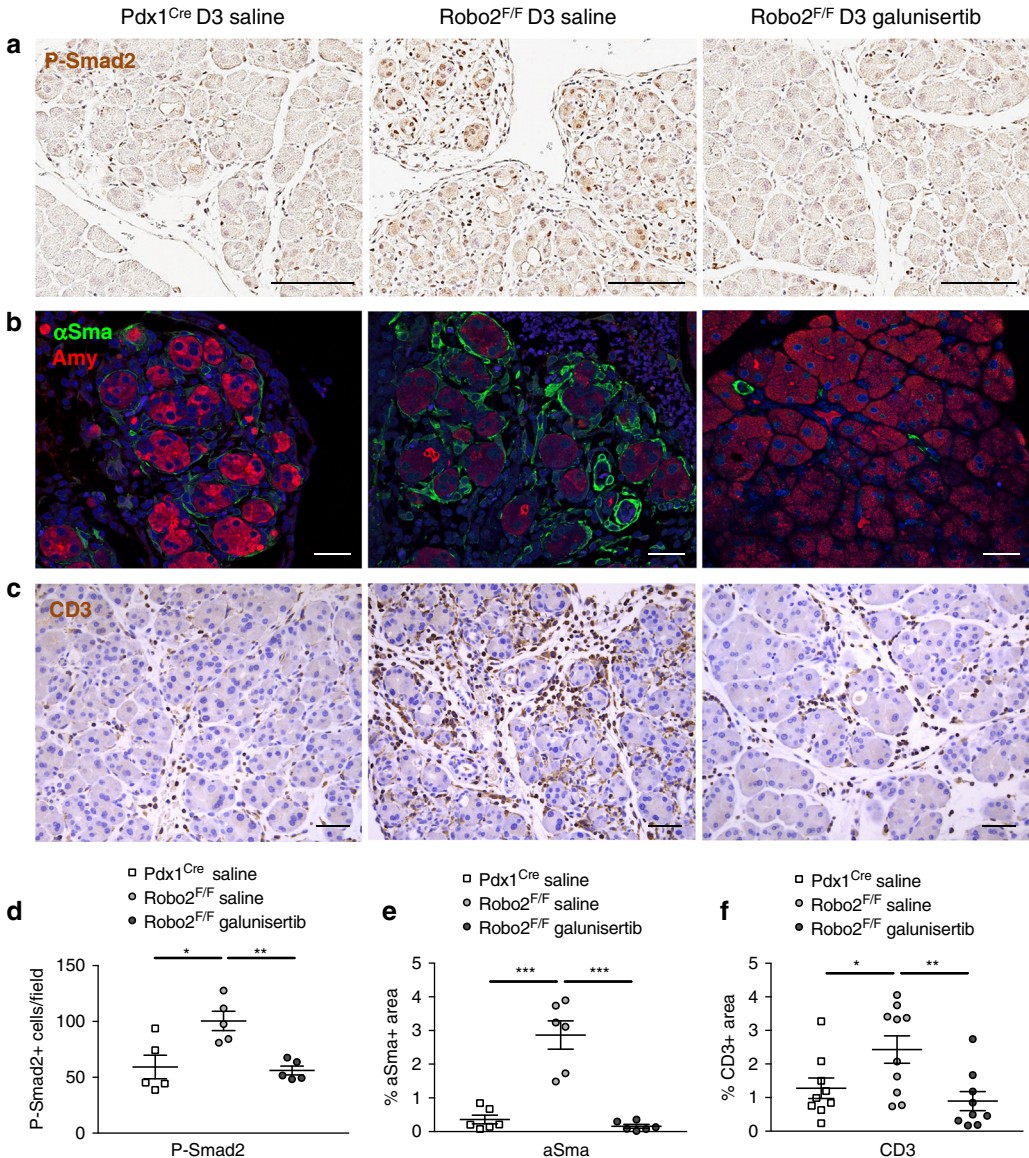

**Fig. 6** TGF-β inhibition suppresses myofibroblast activation and T-cell infiltration. Caerulein-treated animals were simultaneously treated with TGF-β inhibitor galunisertib or saline control by oral gavage. **a** Immunohistochemistry staining of phospho-Smad2 in D3 AP pancreas sections. Nuclei are counterstained with haematoxylin. Images are representative of five independent experiments. Scale bars correspond to 100 μm. **b** Immunofluorescence for the myofibroblast marker αSma and the exocrine marker amylase in D3 AP pancreas sections. Images are representative of five independent experiments. Nuclei are stained with DAPI. Scale bars correspond to 50 μm. **c** Immunohistochemistry staining of the T-cell marker CD3 in D3 AP pancreas sections. Images are representative of five independent experiments. Nuclei are counterstained with haematoxylin. Scale bars correspond to 50 μm. **d** Manual quantification of phospho-Smad2+ cells/field. **e** Quantification of αSma+ area/field, using ImageJ. **f** Quantification of CD3+ area/field, using ImageJ. All images were acquired using 20x magnification. Data presented as mean +/− SEM; N ≥ 5; statistical analysis was performed using an unpaired $t$ test with Welch's correction, $*P < 0.05$, $**P < 0.01$, $***P < 0.001$

**ROBO1 determines prognosis in ROBO2-negative patients**. We investigated the expression of ROBO1 and ROBO2 in a large cohort of PDAC patients ($n = 109$) from APGI[5–7], as well as non-malignant pancreas samples. Similar to the observation in mouse, we found modest expression of ROBO1 and 2 in normal exocrine epithelium and we detected high expression of ROBO1 but low expression of ROBO2 in advanced preneoplastic PanIN lesions (Supplementary Figure 13). The vast majority of PDAC tumours were positive for ROBO1 (106/109) with 55% of patients presenting high expression of ROBO1 (score ≥200 in 60/109 samples). On the contrary, for ROBO2, 54% of PDAC samples were negative (score 0 in 59/109 samples) and 26% only presented low expression (score ≤100 in 28/109 samples) (Fig. 7a, b). Moreover, we found that ROBO1 is also highly expressed by stromal cells in

these tumour samples, while ROBO2 is absent or expressed at very low levels in stromal cells (Fig. 7a).

RNA sequencing (RNAseq) confirmed that ROBO1 was highly expressed in tumours from this cohort, while ROBO2 expression is much lower (Fig. 7c). One should note that RNAseq data were obtained from tumour samples containing epithelial and stromal cells. Tumour epithelial purity, analysed by Qpure analysis[38], negatively correlates with ROBO1 expression in these samples (Spearman $r_s = −0.321$, $p = 0.002$, Fig. 7d), suggesting that ROBO1 mRNA expression is higher in samples with higher stromal content, in agreement with the mouse data. Not only did ROBO1 correlate with abundance of stromal cells, when using the expression signatures from Moffit et al., we found that ROBO1 correlated best with markers of

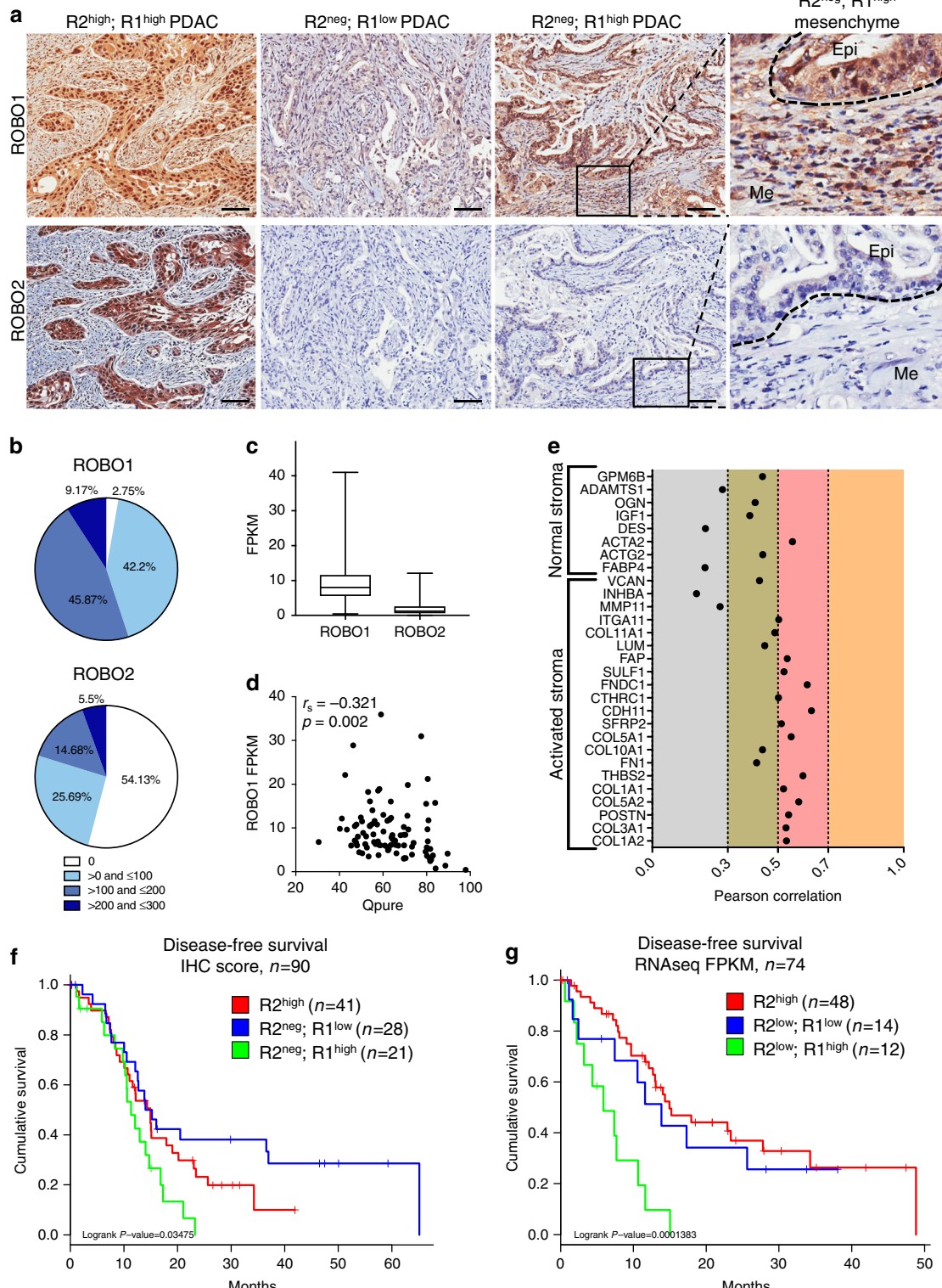

**Fig. 7** ROBO1 determines the prognosis of ROBO2-negative PDAC patients. **a** Representative images of ROBO1 and ROBO2 immunohistochemistry (IHC) in PDAC samples from the APGI cohort. Images were acquired using Aperio ImageScope software at a magnification of 20 × . Scale bars correspond to 50 µm. Dotted lines delineate histological compartments (tumour epithelium, Epi, mesenchymal cells, Me). **b** Graphical representation of ROBO1 and ROBO2 IHC scores in the APGI patient cohort (n = 109). **c** Variation in RNA expression for ROBO1 and 2 within the APGI cohort, analysed by RNAseq and represented as FPKM (RNAseq data previously published by Bailey et al.[7]) (centre line of the box corresponds to mean value, whiskers are the upper and lower limit). **d** Inverse correlation between ROBO1 RNA expression (FPKM) and tumour cellularity analysed using QPure[38]. **e** Correlation between ROBO1 RNA expression (microarray gene expression analysis previously published by Biankin et al.[5]) and markers of normal and activated stroma, as described by Moffitt et al.[39] **f** Kaplan–Meier survival curve with a log-rank statistical test for combined protein expression of ROBO1 and ROBO2, analysed by immunohistochemistry (n = 90). **g** Kaplan–Meier survival curve with a log-rank statistical test for combined RNA expression of ROBO1 and ROBO2, analysed by RNAseq (n = 74)

activated inflammatory stroma (Fig. 7e), which have been associated with poor prognosis[39]. In addition, gene set enrichment analysis of the genes that correlated with ROBO1 in the PDAC samples, was enriched for pathways related to the extracellular matrix, focal and cell adhesion, as well as genes involved in Wnt signalling and TGF-β signalling, reinforcing the findings from the mouse experimental models described above (Supplementary Table 2).

We had previously shown that ROBO2 mRNA expression (from cDNA microarrays) positively correlated with improved patient survival[5]. We now refined our data showing, by immunostaining and by RNAseq, that within the ROBO2[low] population, ROBO1 expression is a determining factor; ROBO2[low];ROBO1[high] tumours corresponding to the patients with the poorest prognosis and ROBO2[low];ROBO1[low] not differing from ROBO2[high] tumours in terms of disease-free survival (Fig. 7f, g).

Altogether, these data illustrate that our functional mouse studies are relevant for human pathology, showing that loss of ROBO2 signalling is commonly found in PDAC and that concurrent high ROBO1 expression correlates with activated Wnt and TGF-β pathways, with markers of activated inflammatory stroma and with poor disease-free survival.

## Discussion

Further to a role in axonal guidance, the SLIT–ROBO pathway has been implicated in cancer[4], with studies showing that Slit2 and Robo1 regulate tumour cell migration and invasion through regulation of the β-catenin/Wnt pathway[11,20–22] and affect neural remodelling in pancreatic cancer[11,12].

Our study reports the involvement of the other partners, Slit1 and Robo2, in regulating the neighbouring microenvironment. We use an integrative approach that combines genetically engineered mice and a large set of human PDAC samples. Specifically, we show that epithelial loss of Robo2 leads to a non-autonomous activation of pancreatic myofibroblasts and induction of an immune response that is primarily of an anti-inflammatory and tumour-promoting nature (Fig. 8). Hence, we coin Robo2 a stroma suppressor gene, a term that we would reserve specifically for a gene expressed in epithelial cells that limits neighbouring stromal cells and that is susceptible to inactivating mutations in cancer.

We not only show that epithelial Robo2 loss is accompanied by expansion of myofibroblasts and induced TGF-β signalling (with prominent phospho-Smad2 in the stromal cells), we also demonstrate dependency on TGF-β signalling. This pathway confers a known stimulus for stroma remodelling, wound healing and desmoplastic response in AP, chronic pancreatitis and PDAC[23,40], but how stroma itself affects cancer development and progression is a matter of vigorous debate[41]. TGF-β itself can also have a dual role herein, depending on the cell type where signalling occurs (epithelium vs. stroma) as well as on the timing (early or late in tumour development); it can function as a tumour suppressor when activated in premalignant epithelial cells, or as a tumour promotor when activated in advanced cancer cells or the tumour microenvironment[42,43]. We demonstrate that Robo2 expression is decreased early in tumour development, in samples with oncogenic Kras activation (mouse KPC model and human PanIN lesions) and is overall low in PDAC samples. Hence, effects of Robo2 loss on cancer development (through TGF-β signalling and stroma activation) might depend on the timing of Robo2 inactivation. Our experiments with concomitant embryonic inactivation of Robo2 and mutant Kras activation (KC_Robo2[F/F]) followed by chronic pancreatitis show no additional phenotypic changes compared to controls. It would be most interesting to sequentially introduce the genetic changes to better mimic human PDAC development.

In our mouse model with epithelial Robo2 deficiency, we also find a non-autonomous activation of Wnt and TGF-β pathways, as well as their common target Robo1[24,25], which is notably activated in stromal cells. Activation of Wnt signalling is actually required for TGF-β-mediated fibrosis[44] and Wnt target genes are markers of activated stroma in the pancreas[39,45]. Indeed, in pancreata of KPC mice, we show expression of Robo1 in αSma[+] myofibroblasts, similar to what is seen in hepatic stellate cells[46,47]. We also demonstrate that in PDAC patients, high ROBO1 mRNA expression inversely correlates with tumour epithelial cellularity and positively correlated with markers of activated stroma, as well

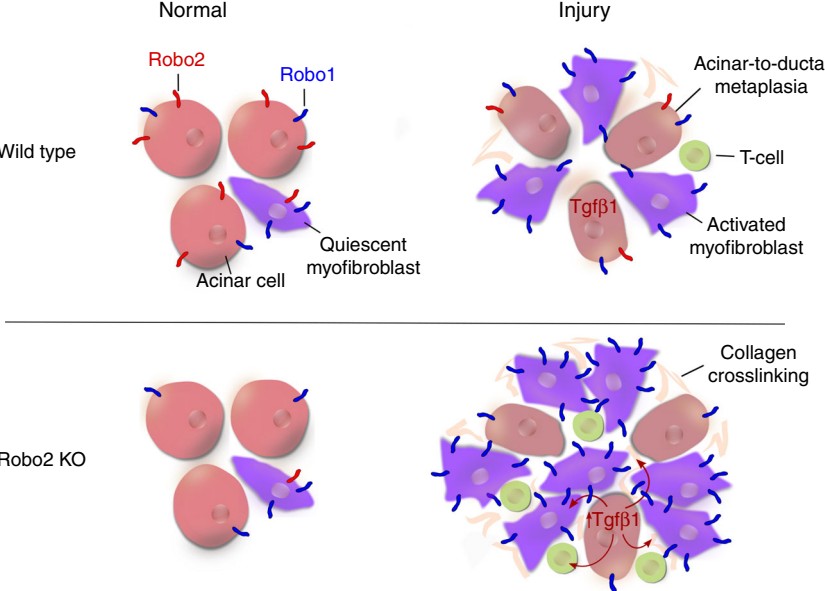

**Fig. 8** ROBO2 is a stroma suppressor gene in the pancreas and acts via TGF-β. A schematic representation of the key findings in this study. After injury, either by cell culture or acute pancreatitis induction, loss of Robo2 in pancreatic epithelial cells leads to non-autonomous activation of pancreatic myofibroblasts and increased T-cell infiltration, dependent on activation of the TGF-β signalling pathway

as with Wnt and TGF-β pathways. High levels of Robo1 in the tumour stroma could on their turn modulate Wnt pathway activation in response to TGF-β activation. We note that Robo1 itself was also reported to activate the Wnt pathway[11,20–22]. The particular mechanisms of this cross-regulation might constitute the subject of future studies. On the other hand, Robo1 is not only found in the myofibroblasts but also frequently found in pancreatic tumour epithelium where, as referred above, it might play a distinct role in migration and invasion[11,20–22].

Pancreata of either Robo2$^{F/F}$ or Slit1$^{-/-}$ have normal organisation of the exocrine and the endocrine tissue. Yang et al. reported that Slit1, 2 or 3 knockdown by siRNA did affect beta-cell survival[48]. We cannot exclude that in our in vivo model, there was a compensatory mechanism at play during development of the pancreas. Indeed, a recent study from Adams et al.[49] suggests that only upon loss of both Robo1 and Robo2 the endocrine compartment is affected.

We had shown that ROBO2 low-expressing tumours present a poorer prognosis[5]. In the present study, we validate these data by RNAseq analyses and demonstrate, using RNAseq and IHC, that within the low ROBO2 population, those patients with high expression of ROBO1 have the poorest prognosis. This led us to propose that epithelial loss of ROBO2 and high expression of ROBO1 may constitute a specific subtype of patients with a characteristic stroma and consequent poor prognosis. The reason why all ROBO2 low patients do not display high ROBO1 expression is still unclear. We speculate that the mutational landscape, e.g. genes within the TGF-β pathway that are frequently affected in PDAC[5,7], might have an influence and other stroma modulators[40] (possibly other stroma suppressor genes) may also have their contribution. With regard to ROBO1 expression in tumour epithelium, it is subject to methylation[8] and hence, different patients may present different expression patterns.

In our preclinical work, Robo2-mediated reprogramming of the microenvironment is blocked by the TGF-β inhibitor galunisertib. TGF-β inhibition has recently been shown to improve the efficacy of immunotherapies in preclinical models of urothelial and colorectal cancer[50,51]. Moreover, phase 2 clinical trials with galunisertib in advanced PDAC showed promising effects on progression-free survival[26,52] and a clinical trial testing galunisertib with immunotherapy is now ongoing[53]. This type of trials, as well as those testing other stroma[54] and/or immune targeting drugs, such as Cxcr2 inhibition[32], could make use of our findings. In conclusion, this report helps to understand the role of the SLIT–ROBO pathway in the pancreatic epithelium and how it regulates the stroma with its fibroblastic as well as immunological components, a topic that is of utmost clinical relevance, given the mutations found in these genes and the drugs that are currently making their way to the clinic.

## Methods

**Animal experimentation**. Experiments were performed in accordance with institutional ethical guidelines and regulations, and were approved by St. Vincent's Hospital Animal Ethics Committee, Sydney, Australia (ethical approval 16/02). In all experiments, adult animals 3–6 months of age and of both sexes were used. Pancreatic-specific Robo2-deficient animals were generated by crossing Robo2 exon 5 flox animals[55] with Pdx1$^{Cre}$ animals[56], resulting in Pdx1$^{Cre}$;Robo2$^{F/F}$ animals. Pdx1$^{Cre}$;Robo2$^{F/F}$ were compared with Pdx1$^{Cre}$ control animals of the same genetic background (129 Sv/C57BL/6). Whole-body Slit1$^{-/-}$ mice[37] were compared with wild-type (WT) animals of the same genetic background (CD-1/Swiss). Pancreatitis was induced by administering 8 hourly intraperitoneal injections of 100 μg/kg of caerulein (Sigma) during 2 consecutive days[13,18]. When applicable, animals were treated with 75 mg/kg of galunisertib (LY2157299, Selleck Chemicals) or saline by oral gavage, twice daily before and after caerulein treatment.

**RNA in situ hybridization**. RNA in situ hybridization (RISH) was performed in formalin-fixed, paraffin-embedded tissue sections, according to the manufacturer's

protocols for manual RNAscope® 2.5 HD Assay—RED (Advanced Cell Diagnostics Inc., 322350). Robo1, Robo2, Slit1, Slit2, Tgfb1, Foxp3, DapB (negative control) and UbC (positive control) probes were purchased from Advanced Cell Diagnostics. For multiplex staining, after RISH, samples were stained with an antibody using routine immunostaining (see below) at 4 °C. Images were acquired using a Nikon Eclipse 90i fluorescence microscope, a Carl Zeiss LSM710 multiphoton confocal laser scanning microscope or an Evos FL1 auto digital microscope (Thermo Fisher Scientific). Image analysis and quantification was done using Image J or Imaris 9.1. (Bitplane). Detailed RISH and imaging protocols are described in the Supplementary Methods.

**Pancreatic exocrine cell isolation and culture**. Pancreatic exocrine cells were isolated and cultured following a method previously published[16,17]. Animals were euthanised in line with animal ethics guidelines and total mouse pancreas was isolated. The pancreatic tissue was digested with a collagenase P (Roche) solution (1.25 mg/mL) for 20 min at 37 °C. Digested pancreata were washed twice in Hank's balanced salt solution (HBSS) (Gibco BRL) supplemented with 5% fetal bovine serum (FBS) (Sigma) and filtered over a 100-μm nylon mesh (Corning). Viable cells were recovered after low-speed centrifugation over 30% FBS. Isolated pancreatic acinar cells were cultured in suspension for the first 48 h in ultra-low-attachment dishes (Corning) to allow dedifferentiation into pancreatic progenitors. From day 3, cells were transferred to tissue culture-treated six-well plates (Corning) and cultured as a monolayer until day 8. Cell culture media consisted of DMEM/F-12 (Gibco) supplemented with 3% FBS, 1X N2 supplement (Invitrogen), 0.5X B27 supplement (Invitrogen), 20 ng/mL EGF (R&D Systems), 100 μM β-mercaptoethanol (Gibco), 1X nonessential amino acids (Gibco), 1X penicillin/streptomycin (Gibco) and 10$^6$ U/L Esgro-LIF (Millipore). Soybean trypsin inhibitor (0.1 mg/mL, Sigma) was added for the first 48 h, and geneticin sulphate (25 μg/mL, Sigma) was added to eliminate contaminating fibroblasts. Galunisertib (5 μM, Selleck Chemicals) or control DMSO was added from the moment of cell isolation and during the 8 days of culture.

**RT-qPCR**. Total RNA was isolated from cells and tissue using Purelink RNA Mini Kit (Ambion, Life Technologies), and cDNA was prepared using Superscript III Reverse Transcriptase (Invitrogen, Life Technologies) according to the manufacturer's instructions. For targeted qPCR, 10 ng of RNA equivalent was used for amplification with specific primers (sequences available upon request) with Power SYBR Master Mix (Invitrogen, Life Technologies) using the 7900 H Fast Real Time PCR System (Life Technologies). All analyses were done in duplicate. A melting curve analysis was performed to control for product quality and specificity. The expression levels were normalised to a housekeeping gene Hprt. Mouse Cancer Inflammation and Immunity Crosstalk RT2 Profiler PCR Arrays (Qiagen) were used for pathway-focused gene expression analysis, following the manufacturer's procedures.

**Flow cytometry**. Pancreatic cell cultures at day 8 were digested using StemCell pro Accutase (Gibco). After washing with DMEM/F-12 (Gibco) with 3% FBS, cells were filtered over a 70-μm nylon mesh. Total mouse pancreas was digested with a collagenase P (Roche) solution (1.25 mg/mL) for 20 min at 37 °C. Digested pancreata were washed in HBSS (Gibco BRL) supplemented with 5% FBS (Sigma) and filtered over a 70-μm nylon mesh (Corning) to obtain a single-cell suspension. Cells were stained with the following fluorophore-conjugated antibodies: Alexa Fluor 488 anti-mouse CD326/EpCAM (dil. 1/50, 118210, Biolegend, Inc.), anti-mouse Cd140a/PDGFR-α (dil. 1/100, 135907, Biolegend, Inc.), PE-Cy7 anti-mouse CD31 (dil. 1/50, 61410, BD Pharmingen) and V450 anti-mouse CD45 (dil. 1/500, 560401, BD Horizon). Propidium iodide (Sigma) was used as death cell exclusion marker. Flow cytometry was performed using FACS Canto II or FACS Aria II (analysis and sorting) from Becton Dickinson and exported to the FlowJo software (Tree Star Inc.) for data analysis.

**Recombination-specific PCR for Robo2$^{F/F}$ animals**. Genomic DNA was extracted from sorted cells using QIAzol Lysis Reagent (Qiagen), according to the manufacturer's conditions. The presence of the Robo2-flox allele was determined by standard PCR amplification of 50 ng of genomic DNA, followed by SpeI (BcuI) restriction digestion with the use of PCR primers Ro2-MEBAC15F and Ro2-MEBAC15R, which amplify a 1100-bp fragment for both wild-type Pdx1-cre and Robo2-flox alleles, as described[55]. After SpeI digestion, the Robo2-flox amplicon remains uncut, whereas the wild-type amplicon yields 750-bp and 350-bp products. Recombination of the Robo2-flox allele was confirmed by amplification of the Robo2del5 allele by primers Robo2koF and Robo2R, which produce a 1180-bp fragment. The Robo2 wild-type allele was amplified by primers Robo2wtF and Robo2R, which yield a 1390-bp fragment. Specific primer sequences are detailed in Supplementary Methods.

**Western blotting**. Proteins were extracted in RIPA buffer supplemented with complete, EDTA-free protease inhibitor cocktail (Roche Diagnostics) and phosphatase inhibitor cocktail 3 (Sigma). Proteins were separated by SDS-PAGE and blotted onto a nitrocellulose membrane. The following antibodies were used: β-actin (dil. 1/5000, A5441, Sigma), phospho-Smad2 (ser465/467) (dil. 1/1000, #3108,

Cell Signalling), Smad2 (dil. 1/1000, #3102, Cell Signalling), E-cadherin (dil. 1/10000, BD610181, BD Pharmingen), α-smooth muscle actin (dil. 1/1000, A5228, Sigma) and vimentin (dil. 1/1000, Ab16700, Abcam). Uncropped blots are presented in the Source data file.

**Immunocytochemistry.** Monolayer-cultured cells were fixed with 4% paraformaldehyde. Antibodies were the following: vimentin (dil. 1/500, AB5733, Millipore), E-cadherin (dil. 1/50, BD610181, BD Pharmingen), α-smooth muscle actin (dil. 1/500, A5228, Sigma) and amylase (dil. 1/100, sc12821, Santa Cruz). After fluorophore-conjugated secondary antibody (Jackson Immunoresearch) incubation, cells were stained with DAPI (Sigma) and mounted using Fluoromount-G (Thermo Fisher Scientific). Immunofluorescence images were acquired with a Leica DMI 6000 SP8 confocal microscope.

Mouse pancreatic tissues were fixed in buffered 4% formaldehyde and embedded in paraffin. Pancreas sections were deparaffinised, hydrated and stained using an automated staining system (Leica BondmaX autostainer). Antigen retrieval was performed at 100 °C with citrate solution (S1699, DAKO). The following antibodies were used: CD3 (dil. 1/50, A0452, DAKO), F4/80 (dil. 1/100, clone Cl:A3-1, AbD Serotec), Krt19/TromaIII (dil. 1/100, Developmental Studies Hybridoma Bank), Axin2 (dil. 1/100, AB32197, Abcam), phospho-Smad2 (dil. 1/10000, gift from Dr. Heldin), insulin (dil. 1/5000) and glucagon (dil. 1/3000) (Diabetes Research Center, Vrije Universiteit Brussel, Brussels, Belgium). EnVision + system (DAKO) was used as a secondary antibody for 30 min before staining with diaminobenzidine (DAB+, DAKO) for 10 min. Immunohistochemistry images were acquired using a Leica DM 4000 microscope.

**Second-harmonic generation imaging.** Second-harmonic generation (SHG) signal derived from H&E sections of pancreata was acquired using a 25 × 0.95 NA water objective on an inverted Leica DMI 6000 SP8 confocal microscope. A Ti: sapphire femtosecond laser cavity was used to excite the samples (Coherent Chameleon Ultra II), operating at 80 MHz and tuned to a wavelength of 840 nm. Intensity was recorded with a RLD HyD detector (420/40 nm). Areas 512 x 512 µm were imaged over a 20-µm z-stack with a z-step size of 1.5 µm.

**Statistical analyses.** Experimental data were analysed by two-tailed unpaired Student $t$ test, unpaired $t$ test with Welch's correction, Mann–Whitney or one-way Anova with Turkey's multiple comparisons test using GraphPad Prism7.0 and statistical significance was accepted at $P < 0.05$. The results are shown as mean ± standard error of mean (SEM). The number of independent experiments ($N$) is indicated in the figure legends.

**PDAC patient samples and clinical data.** Biospecimens, transcriptomic and clinical data were collected by the Australian Pancreatic Cancer Genome Initiative (APGI), and informed consent was obtained from all patients. Use of clinical samples and data were in accordance with national ethical guidelines and regulations in Australia (HREC/11/RPAH/329—Sydney Local Health District—RPA Zone, protocol X11-0220) and in Belgium (UZ Brussels 2017-183, B.U. N.143201732468).

**Patient tissue microarray analysis.** Tissue samples from resected pancreatic ductal adenocarcinomas and clinical data were obtained from patients collected under the auspices of the APGI. Tissues were fixed in 4% formalin and embedded in paraffin. In total, 96 spot tumour microarrays were created using triplicate cores of tumour tissue from each case. Immunohistochemistry (IHC) was performed as described above, using the following antibodies: Robo1 (dil. 1/200, ab7279, Abcam), Robo2 (dil. 1/50, ab75014, Abcam). The intensity of staining is scored using a four-point scale (0, 1, 2 and 3) to represent none, weak, moderate and strong levels of expression. An estimate of the percentage of stained cells was made and multiplied by the intensity score to calculate the $H$-score: $(1 × \% \text{ weak}) + (2 × \% \text{ moderate}) + (3 × \% \text{ strong})$. All sections were scored by the investigator and then second-scored by an expert pathologist without knowledge of patient outcomes. Three cores per tumour were scored, and the median H-score was used. Tumours with median H-score less than 200 were regarded as ROBO1 low-expression group.

**Patient gene expression analysis.** RNA-seq and microarray data from the APGI/International Cancer Genome Consortium (ICGC) pancreatic cancer project have been previously published[5,7], and datasets have been deposited online, as described in the "Data availability" section. The patient tumours were divided into three groups based on ROBO1 and ROBO2 expression levels, determined by RNA-seq. The ROBO1 low- expression group is specified using 45 quantiles as the cut-off, to keep a similar proportion to the low-expression group in IHC data. The correlation between Robo1 gene expression and any other gene on the microarray was tested by both Pearson and Spearman's rank correlation. Genes with R > 4 and R < −4 were selected for pathway enrichment analysis using KOBAS 3.0[57]. The hypergeometrical test was selected to test statistical enrichment of KEGG pathway, and the $P$-values were corrected for multiple comparisons.

**Survival analysis.** Kaplan–Meier survival analysis with a log-rank statistical test was used to analyse disease-free survival from the time of surgery. Patients alive at the time of the follow-up point were censored. All statistical analyses were performed using R/Bioconductor packages.

**Reporting Summary.** Further information on research design is available in the Nature Research Reporting Summary linked to this article.

## Data availability

The authors declare that all data supporting the findings of this study are available within the article and its Supplementary Information files or from the corresponding author upon reasonable request. Microarray data have been deposited at Gene Expression Omnibus (GEO) under the accession code: GSE36924 and RNA-seq data have been deposited in the European Genome-phenome Archive (EGA) under the accession code EGAD00001003298. The source data underlying Figs. 1, 2, 3, 4, 5, 6 and 7 and Supplementary Figs. 2, 3, 4, 5, 6, 7, 8, 9, 10, 11 and 12 are provided as a Source Data file. A reporting summary for this article is available as a Supplementary Information file.

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

## Acknowledgements

Biospecimens and clinical data were provided by the Australian Pancreatic Cancer Genome Initiative (APGI, www.pancreaticcancer.net.au), which is supported by an Avner Pancreatic Cancer Foundation Grant (www.avnersfoundation.org.au). A.V.P. was supported by a fellowship from Cancer Institute NSW, Australia (13/ECF/104). I.R. was supported by the FWO Odysseus programme (Research Foundation Flanders), a fellowship from Cancer Institute NSW (10/FRL/203), Australia and the National Health and Medical Research Council Australia (project 1047343). We thank Maxine Rees, Anaiis Zaratzian, Alice Boulghourjian, Jessica Pettitt, Emmy De Blay, Veerle Laurysens and Geert Stangé for expert technical assistance. We thank Dr. Willem Staels, Vrije Universiteit Brussel, Belgium, and Joao Henriques for the creation of Fig. 8. We thank Dr. Carl-Henrik Heldin, Uppsala University, Sweden for providing the P-Smad2 antibody, Dr. William Andrews, University College London, UK and Dr. Marc Tessier-Lavigne, Rockefeller University, NY, USA, for providing the Robo2flox and Slit1 mouse models, respectively.

## Author contributions

A.V.P. and I.R.: study concept and design, drafting of the manuscript and obtained funding; A.V.P., M.V.B., L.C., M.A., T.S., D.H., C.V. and J.W.: acquisition of data, analysis and interpretation of data, statistical analysis and drafting of the manuscript; D.G.O., AMawson, M.G.L., AMagenau, G.L. and L.B.: acquisition of data, analysis and interpretation of data; A.J.G., P.P., P.T., A.V.B., J.W. and I.R.: analysis and interpretation of data, provided technical and/or material support.

## Additional information

**Competing interests:** The authors declare no competing interests.

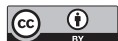

