## [Peer Review File · Nature Communications]

Reviewers' Comments:

Reviewer #1:

Remarks to the Author:

Review of Pinho et al.

Robo2 is a stroma suppressor gene in the pancreas through regulation of TGF- β .

This paper addresses the role of the Robo2 gene in pancreatic cancer using a mouse model. The main findings are noted below with comments immediately following.

The cell type expression of Robo1 and Robo2 is examined by RNA in situ. They conclude that Robo1 is upregulated in stromal cells, while Robo2 is lost in epithelial cells. Fig 1a shows that Robo1 and Robo2 are expressed in acinar cells, but rarely in duct cells.

#1 This point is difficult to see due to what may be compression artefacts in Figure 1a)

They use an ex vivo explant growth model to culture exocrine cell factors (similarly to a method previously described). This model is used with exocrine cells from PDX1CRE-Robo2flox/flox and control PDX1CRE animals. Gene expression changes are shown in Figure 2d,f. Robo2f/f explants show significantly more mesenchymal, stellate cells

#2 The VIM color should be made consistent between panels 2c, 2e

The authors isolated epithelial (EpCAM+) and mesenchymal (Cd140a/Pdgfr-a+) fractions from both types of explants, and show that the mesenchymal cells do not have a floxed Robo2 : from this, they conclude that the mesenchymal fraction is not derived from an epithelial origin. They conclude it must derive from further expansion of existing fibroblastic cells. Since these are not perturbed by Robo2, the Robo2 effect must be non-autonomous.

#3 Their flow cytometry markers should be shown alongside E-Cad, Vim and aSMA (or something subset of these).

#4 The point in italics above seems to be a key element of the paper, but is not explained in sufficient clarity. A cartoon might be helpful.

TGF- β pathway inhibition blocks the effects of epithelial Robo2 loss. The TGF- β pathway is implicated by correlative evidence in Figure 3.

#5 The blot in 3g is not very convincing? (it might be fine given the experimental evidence that follows).

Robo1 is up-regulated, possibly a compensatory mechanism?

#6 Supplementary Figure 4: panel (b) is missing (but described in the legend).

In Figure 4, the authors convincingly show that TGB1 receptor inhibition via galunisertib undoes the gene expression and cellular expansion phenotype seen from Robo2 loss. The authors investigate an Caerulein-treatment acute pancreatitis model, using their Robo2f/f mice. They show stromal expansion (increased interacinar space) at D3.

The authors claim that by D8, the tissue returned to baseline?

#7 I'm not sure this is apparent in Figure 5a. (a relevant 'normal comparison' is in Supp Fig 2a).

They show that, like in vitro, galunisertib treatment could undo the Robo2-associated changes in the context of acute pancreatitis modelling. (Shown in Fig 5). The authors show that Slit1-/- animals recapitulate key features of the Robo2f/f model.

#8 The RNA in situ results in Supp Figure 9 are once again, difficult to interpret.

Finally, the authors show human PDAC can be subclassified using Robo1 and Robo2 expression and that Robo1 expression, in the context of Robo2 loss has a substantially worse prognosis – as predicted by both IHC or RNA-Seq scoring.

A suggestion for the RNA fish images: Figure 1a-c : label the image with dashed lines: endo/acinar/duct?

- Overall, interesting results – impressive use of in vitro modelling to identify an in vivo intervention.

Reviewer #2:

Remarks to the Author:

In this manuscript from Pinho et al, the authors describe a function for ROBO2 as a stroma suppressor gene in the pancreas. The authors show that deletion of Robo2 leads to activation of Robo1 positive myofibroblasts and induction of TGF beta and Wnt pathways. They further present data associating the ROBO2^{low};ROBO1^{high} phenotype as indicative for patients with the poorest survival.

The in vitro data cell culture data are somewhat weak (see points below) and mostly pancreatitis related. Similar the animal model used is an acute pancreatitis model. Hence the authors discuss the role of Robo in context with pancreatic cancer. Therefore, a comparison of KC mice and KC-Robo2F/F mice would significantly strengthen the points suggested in this manuscript.

Figure 1 a-c needs proper quantifications. How many mice per experimental group were analyzed? A quantification of relative expression between acinar cells and ductal cells in NMP, AP and KPC as well as relative increase in expression between NMP, AP and KPC needs to be included.

In Figures 2d, 2f and 3a, 3c the authors compare the pdx1cre with the Robo2F/F culture (similar in 4a-4d). However, the cell types in each culture are quite different (with mainly ADM cells in pdx1cre and mainly fibroblasts in Robo2F/F). The results shown are somewhat trivial, since it is expected that if more fibroblasts are present in one of the culture conditions that the amount of fibroblast markers increases. Is there a difference in the expression of these genes in fibroblasts from either pdx1cre or Robo2F/F (after their separation from other cells in this ADM culture)?

In Fig. 3d and 3f it is unclear if this is an IF picture from RoboF/F. if this is the case, then the control needs to be show, too. In addition, it is unclear if the quantifications shown are comparison of fibroblasts from pdx1cre or Robo2F/F (which would be the proper way to compare the role of switch in Robo isoforms in fibroblast expansion), or analyses of a heterogenous mix of cells.

Figure 4e needs quantifications.

For Figure 6e a table with the fold differences of other genes analyzed needs to be included in the supplemental data.

In Figures 1-6 the authors describe a role of Robo1/2 switching for fibroblast expansion. In the tumor samples in Figure 7a none of the stromal areas express robo1 or 2. Instead both seem to only express in tumor cells. In my opinion this figure is completely disconnected to previous figures.

Other points:

Most of the figure legends are very superficially described. For example in Fig. 3d and 3f it is unclear if this is an IF picture from RoboF/F.

Some of the figures are organized in a different order of how they are discussed in the text, which needs to be changed.

It did not really become clear why the authors call the assay in Fig. 2a an ADM culture with

“exocrine cell fractions with minimal contaminating fibroblasts”. This actually does not look like a typical ADM assay, but rather as a typical assay that allows the outgrowth of fibroblasts. In several figures it is unclear what the size bars indicate.

Reviewer #3:

Remarks to the Author:

Pinho and collaborators have studied the role of Robo1 and Robo2 receptors in pancreatic cancer using a combination of in vitro assay and in vivo models. Their data suggest that Robo1/2 might be involved in PDAC. However, at this stage the work is not a final product, and rather an heterogeneous collection of data, many of which too preliminary. I also think that the ratio of in vitro and in vivo data is not good and that the in vivo ones should be preponderant. Although I read the paper three times and I am still unsure I understood the message. Slit1, 2 and 3 bind similarly to Robo1 and Robo2, so if Robo2 is down and Robo1 is up why is there a problem? The authors should have provided some schematics summarizing the data and working model? How does this work?

To start with : Could the authors present clearly where are Robo1 and Robo2, Slit1, Slit2 and Slit3 are expressed in the normal and pathological pancreas ?

Could they confirm Robo1 and Robo2 expression in acinar cells using immunostaining with commercially available anti-Robo antibodies (several have been validated and some were used for their human studies) ? This also applies to caerulein-induced pancreatitis. The in situ data are not very convincing including the colocalization of Robo2 with SMA+ cells (Fig S1b). Double immunostaining would help.

Slit1, 2 and 3 were all previously detected in pancreatic islets (Yang et al PNAS) which does not really fit with the current data. As Slits function redundantly how removing a single Slit (Slit1) can cause defect? Likewise, the Robo1 KO exist and is viable. It should also be studied to support the conclusions.

They should also show Robo2 expression/level (blot or immuno) in pancreata cultures (control and Robo2^{F/F} ; Figure 2) . What they currently show to support the presence of Robo2 in fibroblastic cells is rather indirect. The absence of Cre expression should obviously be confirmed using Cre immunostaining or preferentially a reporter such as tdTomato.

The authors did not use an inducible line and in Pdx1-cre, and it is known that cre is expressed very early on in development (around E8.5) and therefore some of the observed defects could be related to abnormal pancreas development or cell specification. What supports the normal organization of the pancreas in Pdx1-cre ;Robo2^{lox} mice ? The data supporting normal histology are very weak (only H&E). They should confirm that pancreatic function is normal (RT-PCR is not enough) and quantify the histological data.

A previous study (that the authors fully ignore) showed that Slit/Robo signaling controls beta cell survival (Yang et al., PNAS 110, pp 16480-16485, 2013)

They should then discuss potential discrepancy with this earlier work.

Other points

All the evidence supporting abnormal gene expression in Robo2^{FF} animals with acute pancreatitis comes from RT-PCR (Figure S5).

Figure S4 : why is Robo1 expression not upregulated after Robo2 silencing ?

“Indicative of stromal changes, and an increase of Tgfb1, albeit only a trend (Fig.5b,c).

“a tendency for increased Cd3+ T lymphocyte infiltration at D3”

(S.Fig. 11c,e), "Some typical Wnt target genes showed a tendency for increased expression, albeit not statistically significant"

If some data are not significant this should be clearly stated rather than suggesting that there is a difference. And the picture on the figures should also reflect the results (on S11 it seems that there is a huge increase of CD3 cells)

Could they also validate the specificity of the Robo2 antibody (used for human immunostaining) on Robo2F/f mice ? Were they previously validated? the anti-Robo1 reference ab72790 is an anti DLST based on the abcam site. The Robo2 ab75014 is against the mouse protein and should work on mouse pancreatic tissue.

Why don't they also show Robo1/2 distribution in normal adult human pancreas? It is otherwise difficult to know what Robo2 low or high mean.

Answers to Reviewers' comments

We would like to thank the reviewers for their interest in our work and for their constructive comments and suggestions, as we believe they have led to an improved version of our manuscript. Below we address the particular issues raised by each reviewer.

Reviewer #1:

This paper addresses the role of the Robo2 gene in pancreatic cancer using a mouse model. The main findings are noted below with comments immediately following.

The cell type expression of Robo1 and Robo2 is examined by RNA in situ. They conclude that Robo1 is upregulated in stromal cells, while Robo2 is lost in epithelial cells. Fig 1a shows that Robo1 and Robo2 are expressed in acinar cells, but rarely in duct cells.

#1 This point is difficult to see due to what may be compression artefacts in Figure 1a)

We hope the reviewer can be provided with the non-compressed figures. We have made adjustments in Figure 1 to increase intelligibility and now clearly separate different tissue compartments using a dotted line. In addition, we now provide quantification of RNA expression per cell type for n=6 mice in Fig.1d.

They use an ex vivo explant growth model to culture exocrine cell factors (similarly to a method previously described). This model is used with exocrine cells from PDX1CRE-Robo2flox/flox and control PDX1CRE animals. Gene expression changes are shown in Figure 2d,f. Robo2f/f explants show significantly more mesenchymal, stellate cells

#2 The VIM color should be made consistent between panels 2c, 2e

This change has now been made.

The authors isolated epithelial (EpCAM+) and mesenchymal (Cd140a/Pdgfr-a+) fractions from both types of explants, and show that the mesenchymal cells do not have a floxed Robo2 : from this, they conclude that the mesenchymal fraction is not derived from an epithelial origin. They conclude it must derive from further expansion of existing fibroblastic cells. Since these are not perturbed by Robo2, the Robo2 effect must be non-autonomous.

#3 Their flow cytometry markers should be shown alongside E-Cad, Vim and aSMA (or something subset of these).

These stainings have now been added in supplementary figure 3a, showing that EpCAM co-localizes with ECad and that Cd140a co-localizes with Vimentin. We have also added to the text the reference of a previous study (Ohlund at al. J Exp Med 2017), where the same antibodies have been used to separate the epithelial and mesenchymal cell population in pancreatic tissue.

#4 The point in italics above seems to be a key element of the paper, but is not explained in sufficient clarity. A cartoon might be helpful.

In the correspondence to us there was no text in italics, presumably because of the formatting by the journal. Nevertheless, we have added a cartoon (Fig. 8) that illustrates the key findings of our work.

TGF- β pathway inhibition blocks the effects of epithelial Robo2 loss. The TGF- β pathway is implicated by correlative evidence in Figure 3.

#5 The blot in 3g is not very convincing? (it might be fine given the experimental evidence that follows). \

The Western Blot data of n=6 Pdx1^{Cre} versus n=6 Robo2^{F/F} samples were quantified (Fig. 3d). We acknowledge that there is variability in between the individual experiments which is now visible in the individual data points in the graph. A representative image of n=3 is shown.

Robo1 is up-regulated, possibly a compensatory mechanism?

In supplementary figure 2 we added the RISH analysis of Robo1 expression in Robo2^{F/F} versus controls. We added to the text: 'In the Robo2 depleted epithelial acinar and duct cells, we noticed a slight, and possibly compensatory, upregulation of Robo1 expression (S. Fig. 2b).' Nevertheless, we see a clear increase in Robo1+ mesenchymal cells in both Robo2^{F/F} pancreatic cell cultures and in Robo2^{F/F} animals with caerulein-induced pancreatitis.

#6 Supplementary Figure 4: panel (b) is missing (but described in the legend).

We apologize for this oversight. The legend has now been corrected.

In Figure 4, the authors convincingly show that TGB1 receptor inhibition via galunisertib undoes the gene expression and cellular expansion phenotype seen from Robo2 loss. The authors investigate an Caerulein-treatment acute pancreatitis model, using their Robo2f/f mice. They show stromal expansion (increased interacinar space) at D3. The authors claim that by D8, the tissue returned to baseline?

#7 I'm not sure this is apparent in Figure 5a. (a relevant 'normal comparison' is in Suppl. Fig 2a).

We now put in the text: 'At D8, the epithelial tissue had also regenerated to the same extent as the Pdx1^{Cre} samples (Fig. 5a).' We have included pictures of the control Pdx1^{Cre} samples to support this. We agree that in controls, as well as in ROBO2^{F/F}, the tissue is still not entirely comparable to untreated pancreas. Please also note that this statement referred to the epithelial tissue. 'In contrast, the stromal compartment remained different with increased deposition of collagen (S.Fig. 6c) and increased collagen crosslinking at D8, quantified by second harmonic generation (SHG) microscopy analysis (Fig. 5h,i).'

They show that, like in vitro, galunisertib treatment could undo the Robo2-associated changes in the context of acute pancreatitis modelling. (Shown in Fig 5). The authors show that Slit1-/- animals recapitulate key features of the Robo2f/f model.

#8 The RNA in situ results in Suppl. Figure 9 are once again, difficult to interpret.

Same as for Fig.1, we hope the reviewer can be provided with the uncompressed figures. In addition, we now provide quantification for n=3 mice in Sup. Fig. 10.

Finally, the authors show human PDAC can be subclassified using Robo1 and Robo2 expression and that Robo1 expression, in the context of Robo2 loss has a substantially worse prognosis – as predicted by both IHC or RNA-Seq scoring.

A suggestion for the RNA fish images: Figure 1a-c : label the image with dashed lines

Dashed lines have been added, as suggested by the reviewer.

Overall, interesting results – impressive use of in vitro modelling to identify an in vivo intervention.

We are grateful to the reviewer for the enthusiasm about our findings. We hope that the reviewer is pleased with the way we have now addressed his/her comments.

--

Reviewer #2:

In this manuscript from Pinho et al, the authors describe a function for ROBO2 as a stroma suppressor gene in the pancreas. The authors show that deletion of Robo2 leads to activation of Robo1 positive myofibroblasts and induction of TGF beta and Wnt pathways. They further present data associating the ROBO2low;ROBO1high phenotype as indicative for patients with the poorest survival. The in vitro data cell culture data are somewhat weak (see points below) and mostly pancreatitis related. Similar the animal model used is an acute pancreatitis model. Hence the authors discuss the role of Robo in context with pancreatic cancer. Therefore, a comparison of KC mice and KC-Robo2F/F mice would significantly strengthen the points suggested in this manuscript.

We clarify the rationale for our choice of experimental models. In Fig.1 we show that Robo2 expression is downregulated in acute pancreatitis and in tumor samples of Kras mutant mice (KPC model) epithelial Robo2 expression is lost entirely. Acute pancreatitis and Kras mutant mice represent two conditions of increased Ras signaling.

We confirmed with an additional experiment that overexpression of mutant KRAS^{G12D} in the human pancreatic duct epithelial (HPDE) cell line abolishes ROBO2 expression (see figure below).

Western blot analysis of ROBO1, ROBO2 and Beta-actin protein expression in HPDE cells transfected with mutant KRas or control empty vector.

We first assessed the knockout of *Robo2* on the normal pancreas and did not find a phenotype. Thereafter, we turned to previously published experimental models of ‘prompt’ increased *Ras* signaling in vitro (by cell isolation and culture), and in vivo (by caerulein treatment) (Pinho et al Gut 2011, Logsdon et al. Clin. Gastroenterol. Hepatol. 2009).

Since the KC model has a very strong activation of *Ras* signaling from embryonic development on with concomitant loss of *Robo2* (Fig. 1), we felt that additional *Robo2* loss (by crossing with the *Robo2^{F/F}* line) might not result in a different outcome. This was indeed confirmed, as we show that adult KC pancreas presents a reduced level of *Robo2* expression that is similar to that of the KC or *KC_Robo2^{F/F}* pancreas (See figure below).

RT-qPCR analysis of *Robo2* mRNA expression in adult pancreas of *Pdx1^{Cre}*, *Robo2^{F/F}*, KC and *KC_Robo2^{F/F}* animals. Data is represented as Mean +/- SEM. **p<0.01, ***p<0.001.

Nevertheless, we have now added to our manuscript the analysis of *KC_Robo2^{F/F}* animals treated with caerulein to induce a chronic pancreatitis and accelerate the development of pre-neoplastic lesions. *KC_Robo2^{F/F}* animals presented no differences in PanIN lesion formation, myofibroblast activation or infiltration of CD3⁺ cells. These new data have been provided in Suppl. Fig. 9 and we have incorporated a separate section on this in the main text:

‘Concurrent loss of *Robo2* and *Kras^{G12D}* activation in the pancreas does not lead to enhanced stromal changes after chronic pancreatitis.

As shown in Fig. 1, epithelial *Robo2* expression is lost in different conditions of *Ras* activation, such as AP, where wild-type (WT) *Ras* is overactivated^{17,36} and in the KPC mice¹⁴ that express oncogenic *KRAS^{G12D}*. Loss of *ROBO2* expression was confirmed in human pancreatic duct cells where mutant *KRAS^{G12D}* was overexpressed and in KC animals, which express oncogenic *KRAS^{G12D}* in the pancreas (not shown). We crossed the KC model with *Robo2^{F/F}* animals and treated these animals with caerulein to induce a chronic pancreatitis (S. Fig. 9a). Area occupied by neoplastic lesions was similar between *KC_Robo2^{F/F}* and KC controls, with both groups presenting a mix of early stage lesions (PanIN1a,b and PanIN2) (S. Fig. 9b,c). We also did not find any additional increase in α Sma⁺ cells, CD3⁺ T cell infiltration or higher TGF- β RNA expression (S. Fig. 9d-h).

In conclusion, Robo2 knock-out in the KC mouse background does not increase the typical phenotypic changes in epithelium and stroma. One explanation for this observation is the fact that Kras^{G12D} mutation on its own already induced Robo2 loss.'

In the original manuscript, we had already stated in the discussion, 'it would be most interesting to sequentially introduce the genetic changes to better mimic human PDAC development.' This has now been confirmed. We now elaborate a bit further on this in the discussion:

'We found that Robo2 expression is decreased early in tumor development, in samples with oncogenic Kras activation (mouse KPC model and human PanIN lesions) and is overall low in PDAC samples. Hence effects of Robo2 loss on cancer development (through TGF- β signaling and stroma activation) might depend on the timing of Robo2 inactivation. Our experiments with embryonic inactivation of Robo2, mutant Kras activation (KC; Robo2^{F/F}) followed by chronic pancreatitis showed no additional changes compared to controls (not shown). It would be most interesting to sequentially introduce the genetic changes to better mimic human PDAC development.'

Figure 1 a-c needs proper quantifications. How many mice per experimental group were analyzed? A quantification of relative expression between acinar cells and ductal cells in NMP, AP and KPC as well as relative increase in expression between NMP, AP and KPC needs to be included.

We have added graphs with Robo1 and 2 mRNA quantification for n=6 mice in Fig.1d.

In Figures 2d, 2f and 3a, 3c the authors compare the pdx1cre with the Robo2F/F culture (similar in 4a-4d). However, the cell types in each culture are quite different (with mainly ADM cells in pdx1cre and mainly fibroblasts in Robo2F/F). The results shown are somewhat trivial, since it is expected that if more fibroblasts are present in one of the culture conditions that the amount of fibroblast markers increases. Is there a difference in the expression of these genes in fibroblasts from either pdx1cre or Robo2F/F (after their separation from other cells in this ADM culture)?

In Fig. 3d and 3f it is unclear if this is an IF picture from RoboF/F. if this is the case, then the control needs to be shown, too. In addition, it is unclear if the quantifications shown are comparison of fibroblasts from pdx1cre or Robo2F/F (which would be the proper way to compare the role of switch in Robo isoforms in fibroblast expansion), or analyses of a heterogeneous mix of cells.

We fully agree with the reviewer that the key finding of our paper is the increase of myofibroblasts when comparing the cell cultures from the Robo2^{F/F} to its controls. This was further confirmed now using FACS analysis in Fig. 2g.

Our in vitro model has been developed to study epithelial exocrine cells, hence due to the paucity of (myo)fibroblasts in the control Pdx1^{Cre} cultures, we could not make a comparison of FACS purified fibroblasts from the two conditions. We therefore addressed the question further by quantifying expression using immunostaining and RISH.

The stainings from the original and the reworked Figure 3 are from Robo2^{F/F} cells since this is where we have prominent myofibroblasts. We exactly wanted to make the point that in the PCR analysis from the Robo2^{F/F} condition, the myofibroblasts are the main contributors to the findings. With this figure we did not want to draw the reader's attention to the difference in Pdx1^{Cre} versus Robo2^{F/F}. We rephrased the text alongside: 'The Robo2^{F/F} cell cultures contain a majority of myofibroblasts and these cells stain positive for Axin2, Tgfb1, Robo1 and phospho-Smad2 (P-Smad2) (Fig. 3e), hence they contribute significantly to the increases detected by RT-PCR and underscore the non-autonomous effect of epithelial Robo2 deletion.'

We added in the text that goes with Figure 2: 'We note that both in Pdx1^{Cre} as in Robo2^{F/F} the Vim⁺ cells express α Sma.' Indeed, the few fibroblasts in the Pdx1^{Cre} present expression of the same markers and Robo2^{F/F} mesenchymal cells (See figure below, for reviewer only).

Pdx1^{Cre} Mesenchymal cells

Immunofluorescence/RISH of Axin2, Robo1, Tgfb1 and phospho-Smad2 mesenchymal cells from Pdx1^{Cre} cultures.

In addition, we quantified the expression of α Sma and Robo1 per vim⁺ cell. This was incorporated in Fig. 4 so that the repression by galunisertib can be appreciated. 'We observed that in the mesenchymal cells of the Robo2^{F/F} cultures, the expression of α Sma as well as that of Robo1 was suppressed by galunisertib (Fig. 4e,f).'

We would not call our observation 'trivial' since it is a unique phenotype that we describe in this manuscript. It is an important finding because of Robo2 knock-out being specific for the epithelial cells while the key outcome is the myofibroblast activation.

Figure 4e needs quantifications.

Same as above, we quantified the expression of α Sma and Robo1 per vim⁺ cell. This was incorporated in Fig. 4 so that the repression by galunisertib can be appreciated. 'We observed that in the mesenchymal cells of the Robo2^{F/F} cultures, the expression of Acta2 as well as that of Robo1 was suppressed by galunisertib (Fig. 4e,f).'

For Figure 6e a table with the fold differences of other genes analyzed needs to be included in the supplemental data.

This has now been included as Supplementary Table 1.

In Figures 1-6 the authors describe a role of Robo1/2 switching for fibroblast expansion. In the tumor samples in Figure 7a none of the stromal areas express robo1 or 2. Instead both seem to only express in tumor cells. In my opinion this figure is completely disconnected to previous figures.

Staining for ROBO1 can be observed both in tumor epithelium and stromal cells. We have now added dashed lines on the inset pictures to clearly demarcate tumor epithelium and stroma.

Other points:
Most of the figure legends are very superficially described. For example in Fig. 3d and 3f it is unclear if this is an IF picture from RoboF/F.

Some of the figures are organized in a different order of how they are discussed in the text, which needs to be changed.

It did not really become clear why the authors call the assay in Fig. 2a an ADM culture with “exocrine cell fractions with minimal contaminating fibroblasts”. This actually does not look like a typical ADM assay, but rather as a typical assay that allows the outgrowth of fibroblasts. In several figures it is unclear what the size bars indicate.

The figure order has been modified as much as possible according to text and the figure legends have been extended. The text has been adapted alongside.

We rephrased the introduction of the cell culture model and added more references. ‘An established in vitro assay with highly enriched exocrine cell fractions mimics the epithelial cell changes (acinar to ductal metaplasia) that occur during pancreatitis¹⁵⁻¹⁸. This is a published assay in which the exocrine cells undergo acinar to ductal metaplasia that was used in several publications and has been recapitulated even with human cells (references 15-18). It is not an assay meant to have fibroblast outgrowth. We actually never saw substantial fibroblast outgrowth with any of the genetically engineered mice that we cultured the epithelial cells from. This is why the observations in the Robo2^{F/F} were so striking.

--

Reviewer #3 (Remarks to the Author):

Pinho and collaborators have studied the role of Robo1 and Robo2 receptors in pancreatic cancer using a combination of in vitro assay and in vivo models. Their data suggest that Robo1/2 might be involved in PDAC. However, at this stage the work is not a final product, and rather an heterogeneous collection of data, many of which too preliminary. I also think that the ratio of in vitro and in vivo data is not good and that the in vivo ones should be preponderant. Although I read the paper three times and I am still unsure I understood the message. Slit1, 2 and 3 bind similarly to Robo1 and Robo2, so if Robo2 is down and Robo1 is up why is there a problem? The authors should have provided some schematics summarizing the data and working model? How does this work?

It is unsettling to find out that we failed to convey the message of our findings to the reviewer. In the revised version of the manuscript, we introduced a schematic (Fig. 8) that may assist in conveying the key message being ‘loss of Robo2 in pancreatic epithelial cells leads to a non-autonomous activation of pancreatic myofibroblasts and immune cells that is dependent of the TGF-beta pathway’.

We respectfully disagree with the reviewer that there is a preponderance of in vitro data. We have 3 main figures on in vitro experimental analysis, 2 on in vivo experimental analysis and 1 on patient sample analysis. In addition, there are 4 supplementary figures displaying in vitro data versus 9 supplementary figures with in vivo data.

We consciously showed a mix of experimental models to substantiate our findings (which was highly appreciated by reviewer 1). All results show that epithelial Robo2 expression decreases during pancreatitis and even more in cancer, two conditions of activated Ras signaling. We started off with a simple model of cell culture where a prompt Ras activation occurs (Pinho et al, Gut 2011). Because of the apparent phenotype, we validated these findings in vivo in the caerulein model. Since stromal cell activation is mainly a topic of investigation in pancreatic cancer, we included an analysis of one of the largest datasets of human pancreatic cancer samples with clinical annotation. Following the request from reviewer 2, we have now also added data of the KC mouse model of pancreatic cancer (Suppl. Fig. 9), albeit that the timing of the genetic changes in this model is not adequate, as we anticipated, and hence no difference between Robo2^{F/F} and controls was found. Instead of restarting long term breeding experiments with more suitable mouse models, we felt that the human data were sufficiently supporting our findings that they could be added in without being perceived as too disperse.

To start with : Could the authors present clearly where are Robo1 and Robo2, Slit1, Slit2 and Slit3 are expressed in the normal and pathological pancreas ?

We now provide in Fig.1 and Suppl. Fig. 10 a quantification of RNA expression analysis in acinar cells, duct cells and mesenchymal cells in normal and pathological pancreas (in both acute pancreatitis and PDAC models). Islets, which are outside the scope of our study, are shown for illustration in Suppl. Fig. 2.

Could they confirm Robo1 and Robo2 expression in acinar cells using immunostaining with commercially available anti-Robo antibodies (several have been validated and some were used for their human studies) ? This also applies to caerulein-induced pancreatitis. The in situ data are not very convincing including the colocalization of Robo2 with SMA+ cells (Fig S1b). Double immuno would help.

The signals retrieved by RNAScope are very specific, inherent to the probe design. Antibodies in general are more prone to unspecificity. The use of several antibodies for Robo1 and Robo2 has been reported in literature. The antibodies from Santa Cruz, used in Yang et al (PNAS 2013), have been discontinued by the company. We refrained from using the anti-ROBO2 from Sigma since it results in a nuclear staining, which we are unsure is specific and it detects a 37Da band whereas the predicted molecular weight is 175 kDa. (<https://www.proteinatlas.org/ENSG00000185008-ROBO2/antibody>).

For our human studies, we used antibodies from ABCAM, whose specificity was confirmed in cells with overexpression of either human ROBO1 or ROBO2 (see below, for reviewer only).

HEK293T cells were transiently reverse-transfected with full length c-terminally tagged ROBO1 (Flag) or ROBO2 (Myc) expression vectors with lipofectamine 2000 (100000 cells/well, 24 well-plate format) according to the manufacturer's instructions. Empty vector was used as a negative control. 48 hours post-transfection cells were lysed and subjected to western blot analysis.

Left panel: The observations for these analyses confirm specificity of the ROBO1 and ROBO2 antibodies. Anti-ROBO1 antibody detects low levels of endogenous ROBO1 protein as well (see long exposure).

Right panel: Separate membranes loaded with HEK293T ROBO1 and ROBO2 transfectants were incubated with ROBO1 and ROBO2 antibodies. ROBO1 and ROBO2 antibodies did not exhibit cross-reactivity for ROBO2 and ROBO1 proteins.

Nevertheless, the same antibody for Robo2 failed to work on our mouse samples used throughout the manuscript, i.e. Western Blot did not show any bands around the reported molecular weight of approx. 175 kDa whereas the human ROBO2 protein could be easily detected (see first figure below, for reviewer only). Also, in the mouse 266-6 acinar cell line, no band could be detected at the predicted molecular weight (second figure below, panel c, for reviewer only) in contrast to Robo2 detection by RT-PCR in these samples (panel a, for reviewer only) or by RISH (panel b, for reviewer only).

Western blotting using Robo2 antibodies directed for the C-terminal and N-terminal regions of Robo2 in primary pancreatic cultures of Pdx1^{Cre} and Robo2^{F/F} animals.

a

	average Ct Hprt	average Ct Robo2
266-6 cells sample1	24,41137233	32,306244
266-6 cells sample2	23,98909567	31,97604433
266-6 cells sample3	24,20904533	32,04860067

b

c

Analysis of Robo2 expression in the mouse acinar cell line 266-6. **a.** Ct values from real time RT-PCR analysis of mouse acinar cell line 266-6. Hprt was used as housekeeping gene. **b.** Representative image of Immunofluorescence staining for E-cad and Robo2 mRNA in 266-6 cell line. Nuclei are stained with Dapi. Images were acquired at 20x magnification. Scale bar corresponds to 50 μ m. **c.** Western blotting using Robo2 antibody ab85278 in total cell lysates of 266-6 cell line.

Slit1, 2 and 3 were all previously detected in pancreatic islets (Yang et al PNAS) which does not really fit with the current data. As Slits function redundantly how removing a single Slit (Slit1) can cause defect? Likewise, the Robo1 KO exist and is viable. It should also be studied to support the conclusions.

Our study was initiated because of the genomic aberrations found in PDAC, a cancer of the exocrine pancreas, and more specifically a loss of ROBO2 that conferred poor survival outcomes in PDAC patients (Biankin et al. Nature 2012). Hence, our experimental work focused on experimental models of the exocrine pancreas and on deletion of Robo2.

However, we do report the expression of Slits in pancreatic islets (Supp. Fig.10) where we see cells positive for Slit1 and Slit2 in the islet periphery. This result is in discrepancy with the reported expression by Yang et al. in the majority of islet cells. Moreover, where our Pdx1^{Cre} line also targets deletion of Robo2 to the pancreatic islets, we did not see an apparent phenotype (i.e. islet morphology was normal (see also pictures in Supp. Fig. 2 and below). The latter is in contrast with Yang et al who, using siRNA in islet cells, reported an effect on islet viability. One plausible explanation is that the genetic deletion from early embryonic development in our model has been compensated for.

Representative images of immunofluorescence staining for Insulin, Glucagon and Amylase showing no difference between $Pdx1^{Cre}$ and $Robo2^{F/F}$. Blood shows unspecific fluorescence in the green channel. Images acquired using 10x magnification for overview purposes. High magnification is used in S.Fig. 2d.

*Slits are suggested to have largely redundant functions but this does not exclude specialized functions of these ligands*¹. As mentioned in the present study, *Slit2* deficient mice do not survive while *Slit1* deficient mice do, the reason why we proceeded with studying the *Slit1* deficient mouse model. Indeed, this distinctive lethality of *Slit2* knockout versus *Slit1* knockout suggests some degree of non-redundant and specialized functions of *Slit1* and *Slit2* in mouse. In the literature, other examples of such specialized functions of *Slits* can be found. For instance, *Slit2* knockout but not *Slit1* or *Slit3* knockout will result in defasciculation phenotype in mice². In another context, despite the fact that both *Slit 1* and *Slit2* are cooperating during development of mouse visual system, still a specialized role has been described for *Slit2* during retinal neovascularization³. Additionally as we learn from Yang et.al⁴, *Slit3* alone but not *Slit1* or *Slit2* individually, could rescue a phenotype involving pancreatic islet cell death, another indication that specialized non-redundant function of *Slits* do exist.

1. Blockus, H. & Chedotal, A. *Slit-Robo signaling*. *Development* 143, 3037–3044 (2016).
2. Jaworski, A. & Tessier-Lavigne, M. *Autocrine/juxtacrine regulation of axon fasciculation by Slit-Robo signaling*. *Nat. Neurosci.* 15, 367 (2012).
3. Rama, N. et al. *Slit2 signaling through Robo1 and Robo2 is required for retinal neovascularization*. *Nat. Med.* 21, 483–491 (2015).
4. Yang, Y. H. C., Manning Fox, J. E., Zhang, K. L., MacDonald, P. E. & Johnson, J. D. *Intraislet SLIT-ROBO signaling is required for beta-cell survival and potentiates insulin secretion*. *Proc. Natl. Acad. Sci. U. S. A.* 110, 16480–16485 (2013).

Initially, we did enquire to obtain Robo1 knock out mice for our study but we received a polite email that these mice could not be made available to us (see copy below). Knowing our results now, we would have to delete Robo1 expression in the fibroblast lineage to complement our current observations. We believe that for the key message of the current manuscript there is no need to engage in that study.

Extract from email correspondence:

27-apr-2012 from Andrews, Bill

Dear ilse,

XXXX While I can allow access to the Robo2 mouse to study [REDACTED] I cannot allow access to Robo1 as [REDACTED] and I think there would be some conflict of interest there.

Let me know how you want to proceed!

Best wishes

Bill

They should also show Robo2 expression/level (blot or immuno) in pancreata cultures (control and Robo2F/F ; Figure 2) . What they currently show to support the presence of Robo2 in fibroblastic cells is rather indirect. The absence of Cre expression should obviously be confirmed using cre immuno of preferentially a reported such as tdTomato.

As stated above, we failed to find a reliable antibody for detection of Robo2 in our mouse samples.

We are somewhat confused about the reviewer's comment. At no point, we aim to show Robo2 presence in fibroblasts. We aim to show that the fibroblasts are not derived by EMT from the Robo2-depleted epithelium. As suggested by the reviewer, we performed a staining for Cre. Cre is not stained in the fibroblasts whereas it does in the epithelial cells (figure below, for the reviewer only). However, this is not a conclusive result since the Cre expression is driven by Pdx1 and Pdx1 is not expected to be expressed by mesenchymal cells. The only conclusive result can come from a lineage labelling experiment, as suggested by the reviewer, or from detecting the recombination event by genomic DNA analysis as shown in Figure 2h.

Representative confocal images of immunofluorescence staining for α Sma and Cre. Cre is present only in α Sma⁺ cells. Nuclei are stained with Dapi. Images acquired using 20x magnification. Scale bars correspond to 50 μ m.

The authors did not use an inducible line and in Pdx1-cre, and it is known that cre is expressed very early on in development (around E8.5) and therefore some of the observed defects could be related to abnormal pancreas development or cell specification. What supports the normal organization of the pancreas in Pdx1-cre ;Robo2lox mice ? The data supporting normal histology are very weak (only H&E). They should confirm that pancreatic function is normal (RTPCR is not enough) and quantify the histological data.

In line with a recent report by Adams et al (Sci Rep 2018, 8: 10876) we did not see any abnormal pancreatic histology in the single KO of Robo2 or Slit1, i.e. acini presented normal organization, endocrine islets were nicely dispersed throughout the tissue with peripherally located glucagon-positive cells (Suppl. Fig.2), ducts presented normal staining pattern for duct cell marker CK19 (not shown). Staining for amylase in basal conditions did not reveal any difference between Robo2^{F/F} and controls (Suppl. Fig.2). Ageing was normal and mice culled at 6 months for Robo2^{F/F} or for Slit1KO did not display signs of metabolic distress.

A previous study (that the authors fully ignore) showed that Slit/Robo signaling controls beta cell survival (Yang et al., PNAS 110, pp 16480-16485, 2013) They should then discuss potential discrepancy with this earlier work.

In the discussion we now introduced “pancreata of either Robo2^{F/F} or Slit1KO had normal organisation of the exocrine and the endocrine tissue. Yang et al reported that Slit1,2 or 3 knock down by siRNA did affect beta-cell survival. We cannot exclude that in our in vivo model there was a compensatory mechanism at play during development of the pancreas. Indeed, the study from Adams et al. suggest that only upon loss of Robo1 and Robo2 the endocrine compartment is affected”.

Other points

All the evidence supporting abnormal gene expression in Robo2^{FF} animals with acute pancreatitis comes from RTPCR (Figure S5).

We agree that most data come from RT-PCR. However, Figs 5, 6 and Supp. Fig S8 show data by immunostaining and RNAScope ISH in acute pancreatitis in the Robo2^{F/F} line.

Figure S4 : why is Robo1 expression not upregulated after Robo2 silencing ?

We have detected an increase in expression of Robo1 in epithelial cells when Robo2 is silenced (S. Fig.2 a, b). The major increase in Robo1 that we see in our mixed cultures comes however from the other cell type, i.e. the fibroblasts (Fig 4 d, f).

"Indicative of stromal changes, and an increase of Tgfb1, albeit only a trend (Fig.5b,c). "a tendency for increased Cd3+ T lymphocyte infiltration at D3"(S.Fig. 11c,e), "Some typical Wnt target genes showed a tendency for increased expression, albeit not statistically significant"

If some data are not significant this should be clearly stated rather than suggesting that there is a difference. And the picture on the figures should also reflect the results (on S11 it seems that there is a huge increase of CD3 cells)

We tried to be as detailed and correct as possible when describing the results. The dotted graphs with the individual data points are now better reflecting the variability that often is encountered when doing in vivo analyses.

Could they also validate the specificity of the Robo2 antibody (used for human immunostaining) on Robo2^{F/f} mice ? Were they previously validated? the anti-Robo1 reference ab72790 is an anti DLST based on the abcam site. The Robo2 ab75014 is against the mouse protein and should work on mouse pancreatic tissue. Why don't they also show Robo1/2 distribution in normal adult human pancreas? It is otherwise difficult to know what Robo2 low or high mean.

We apologize for the typo when referring to the antibody catalogue number. It should be ab7279 instead of ab72790. We would like to refer the reviewer to Suppl. Fig. 12 where we did show the staining on normal human pancreas. The validity of the human antibodies is shown above.

Reviewers' Comments:

Reviewer #1:

Remarks to the Author:

Comments on Nat Comm 162910-1

“ROBO2 is a stroma suppressor gene in the pancreas through regulation of TGF- β ”

Pinho et al.

The authors response to the various criticisms and suggestions is thorough and largely complete.

General comments:

- Fig 1 is much improved and now more convincing.
- Quantification of the Western blot (Fig3d) is an improvement.
- Some of the effects of galunisertib are evident, but on the whole modest, given the high variation in the quantification. This does not detract from the utility of these experiments (Figs 4 &6), but the conclusions might be stated with a bit more constraint (given legitimate concerns about inhibitor specificity and efficacy).
- Changes to the order of the figure panels and the accompanying text has improved the presentation.
- addition of the summary figure (8) is helpful.

Overall, the authors have attended to the 3 reviewers comments with satisfactory results. The authors are to be commended for taking the requests so seriously and I find their responses helpful and complete.

Reviewer #2:

Remarks to the Author:

All my points have been addressed sufficiently.

Reviewer #3:

Remarks to the Author:

The authors have done their best to address my concerns and I am satisfied with the revision although the underlying mechanism is not entirely clear.

ROBO2 is a stroma suppressor gene in the pancreas and acts via TGF- β signalling
Pinho et al.

RESPONSE TO REVIEWERS' COMMENTS:

We would like to thank all reviewers for their kind comments to our reviewed manuscript and for having acknowledged the effort made by our team to answer all their initial comments. We believe the review has led to an improved version of our initial study.

Please see answers to specific reviewers below, when applicable.

Reviewer #1 (Remarks to the Author):

Comments on Nat Comm 162910-1

“ROBO2 is a stroma suppressor gene in the pancreas through regulation of TGF- β ”

Pinho et al.

The authors response to the various criticisms and suggestions is thorough and largely complete.

General comments:

- Fig 1 is much improved and now more convincing.
- Quantification of the Western blot (Fig3d) is an improvement.
- Some of the effects of galunisertib are evident, but on the whole modest, given the high variation in the quantification. This does not detract from the utility of these experiments (Figs 4 &6), but the conclusions might be stated with a bit more constraint (given legitimate concerns about inhibitor specificity and efficacy).

We have now rephrased the text of sections referring to figures 4 and 6 and stated our conclusions in a more constrained manner, as suggested:

“In Robo2F/F cell cultures treated with galunisertib, there was an inhibition of the effects seen in untreated Robo2F/F cells, i.e. epithelial markers E-cad, Krt19 and Cpa1 were upregulated, and mesenchymal markers Vim, Snail, Acta2/ α Sma, as well as Tgfb1, Tgfbr2 or Robo1 were downregulated, compared with Robo2F/F controls (Fig. 4a-c). Changes in α Sma, Vim and E-cad were confirmed by western blot (Supplementary Figure 4b) and immunofluorescence (Fig. 4d). We observed that in the mesenchymal cells of the Robo2F/F cultures, the expression of α Sma as well as that of Robo1 was suppressed by galunisertib (Fig. 4e,f). In addition, we found that galunisertib reduced the Tgfb1 increase found in Robo2F/F epithelial cells (Supplementary Figure 5a,b).”

“Galunisertib decreased the accumulation of α Sma+ myfibroblasts in Robo2F/F (Fig. 6b,e) as well as the infiltration of CD3+ T lymphocytes (Fig. 6c,f).”

- Changes to the order of the figure panels and the accompanying text has improved the presentation.
- addition of the summary figure (8) is helpful.

Overall, the authors have attended to the 3 reviewers comments with satisfactory results. The authors are to be commended for taking the requests so seriously and I find their responses helpful and complete.

Reviewer #2 (Remarks to the Author):

All my points have been addressed sufficiently.

Reviewer #3 (Remarks to the Author):

The authors have done their best to address my concerns and I am satisfied with the revision although the underlying mechanism is not entirely clear.

We acknowledge that further work is needed to completely understand the role of this complex pathway and the interaction between the different cell types. Research in our group will further dissect the mechanisms of Slit_Robo signalling in the context of pancreatic cancer.